# Large-width asymptotics and training dynamics of $\alpha$-Stable ReLU neural networks

**Stefano Favaro**                                                     *stefano.favaro@unito.it*
*Department of Economics and Statistics*
*University of Torino and Collegio Carlo Alberto*

**Sandra Fortini**                                                 *sandra.fortini@unibocconi.it*
*Department of Decision Sciences*
*Bocconi University*

**Stefano Peluchetti**                                                   *phd.st.p@gmail.com*
*Cogent Labs, Tokyo*

**Reviewed on OpenReview:** *https: // openreview. net/ forum? id= bEwAAEmRbh*

## Abstract

Large-width asymptotic properties of neural networks (NNs) with Gaussian distributed weights have been extensively investigated in the literature, with major results characterizing their large-width asymptotic behavior in terms of Gaussian processes and their large-width training dynamics in terms of the neural tangent kernel (NTK). In this paper, we study large-width asymptotics and training dynamics of $\alpha$-Stable ReLU-NNs, namely NNs with ReLU activation function and $\alpha$-Stable distributed weights, with $\alpha \in (0, 2)$. For $\alpha \in (0, 2]$, $\alpha$-Stable distributions form a broad class of heavy tails distributions, with the special case $\alpha = 2$ corresponding to the Gaussian distribution. Firstly, we show that if the NN's width goes to infinity, then a rescaled $\alpha$-Stable ReLU-NN converges weakly (in distribution) to an $\alpha$-Stable process, which generalizes the Gaussian process. As a difference with respect to the Gaussian setting, our result shows that the activation function affects the scaling of the $\alpha$-Stable NN; more precisely, in order to achieve the infinite-width $\alpha$-Stable process, the ReLU activation requires an additional logarithmic term in the scaling with respect to sub-linear activations. Secondly, we characterize the large-width training dynamics of $\alpha$-Stable ReLU-NNs in terms an infinite-width random kernel, which is referred to as the $\alpha$-Stable NTK, and we show that the gradient descent achieves zero training error at linear rate, for a sufficiently large width, with high probability. Differently from the NTK arising in the Gaussian setting, the $\alpha$-Stable NTK is a random kernel; more precisely, the randomness of the $\alpha$-Stable ReLU-NN at initialization does not vanish in the large-width training dynamics.

## 1 Introduction

There exists a vast literature on the interplay between Gaussian processes and the large-width asymptotic behaviour of Gaussian neural networks (NNs), namely NNs with Gaussian distributed weights (Neal, 1996; Der and Lee, 2006; Garriga-Alonso et al., 2018; Lee et al., 2018; Matthews et al., 2018; Novak et al., 2018; Yang, 2019;a;b; Bracale et al., 2021; Eldan et al., 2021; Klukowski, 2022; Yang and Hu, 2021; Basteri and Trevisan, 2022; Favaro et al., 2023; Hanin, 2023; Trevisan, 2023; Hanin, 2024). To define a Gaussian NN, consider the following elements: i) for $d, k \geq 1$ let $X$ be a $d \times k$ NN's input, such that $x_j = (x_{j1}, \ldots, x_{jd})^T$ is the $j$-th input (column vector); ii) let $\phi$ be an activation function; iii) for $m \geq 1$ let $W = (w_1^{(0)}, \ldots, w_m^{(0)}, w)$ be the NN's weights, such that $w_i^{(0)} = (w_{i1}^{(0)}, \ldots, w_{id}^{(0)})$ and $w = (w_1, \ldots, w_m)$ with the $w_{ij}^{(0)}$'s and the $w_i$'s

being i.i.d. according to a Gaussian distribution with mean 0 and variance $\sigma^2$. A Gaussian $\phi$-NN of width $m$ is

$$f_m(W, X, \phi) = (f_m(W, x_1, \phi), \ldots, f_m(W, x_k, \phi)), \tag{1}$$

where

$$f_m(W, x_j, \phi) = \sum_{i=1}^{m} w_i \phi(\langle w_i^{(0)}, x_j \rangle) \qquad j = 1, \ldots, k.$$

Neal (1996) first investigated the large-width behaviour of $f_m(W, X, \phi)$, which follows by a straightforward application of the central limit theorem (CLT). In particular, it is well-known that if $m \to +\infty$, then the rescaled Gaussian $\phi$-NN $m^{-1/2} f_m(W, X, \phi)$ converges weakly (or in distribution) to a Gaussian process with covariance function $\Sigma_{X,\phi}$ such that $\Sigma_{X,\phi}[r, s] = \sigma^2 \mathbb{E}[\phi(\langle w_i^{(0)}, x_r \rangle) \phi(\langle w_i^{(0)}, x_s \rangle)]$. Some extensions of this infinite-width limit are available for deep NNs (Matthews et al., 2018), more general NN's architectures (Yang, 2019a;b), and infinite-dimensional inputs (Bracale et al., 2021; Eldan et al., 2021; Favaro et al., 2023).

The large-width training dynamics of Gaussian NNs has been also extensively investigated in the literature, with the training being performed through the gradient descent (Jacot et al., 2018; Arora et al., 2019; Du et al., 2019; Lee et al., 2019). In particular, consider the Gaussian ReLU-NN $f_m(W, X) = f_m(W, X, \text{ReLU})$, and set

$$\tilde{f}_m(W, X) := \frac{1}{m^{1/2}} f_m(W, X).$$

Let $(X, Y)$ be the training set, where $Y = (y_1, \ldots, y_k)$ is the (training) output such that $y_j$ corresponds to the $j$-th input $x_j$. By considering a random initialization $W(0)$ for the NN's weights, and assuming a squared-error loss, the gradient flow of $W(t)$ leads to the training dynamics of $\tilde{f}_m(W(t), X)$, that is for any $t \geq 0$

$$\frac{\mathrm{d}\tilde{f}_m(W(t), X)}{\mathrm{d}t} = -(\tilde{f}_m(W(t), X) - Y) \eta_m H_m(W(t), X), \tag{2}$$

where $\eta_m > 0$ is the learning rate, and $H_m(W(t), X)$ is a $k \times k$ random matrix whose $(j, j')$ entry is $\langle \partial \tilde{f}_m(W(t), x_j)/\partial W, \partial \tilde{f}_m(W(t), x_{j'})/\partial W \rangle$. By assuming $\eta_m = 1$, Jacot et al. (2018) first characterized the large-width training dynamics of $\tilde{f}_m(W(t), X)$, showing that: i) if $m \to +\infty$ then $H_m(W(0), X)$ converges in probability to a deterministic matrix $H^*(X, X)$; ii) the gradient descent achieves zero training error at linear rate, i.e.

$$\|Y - \tilde{f}_m(W(t), X)\|_2^2 \leq \exp(-\lambda_0 t) \|Y - \tilde{f}_m(W(0), X)\|_2^2$$

for $m$ sufficiently large, with high probability. The limiting matrix $H^*(X, X)$ is refereed to as the neural tangent kernel (NTK). See Yang (2019) and Yang and Littwin (2021) for extensions to deep NNs and general architectures.

## 1.1 Our contributions

In this paper, we study large-width asymptotics and training dynamics of $\alpha$-Stable ReLU-NNs, namely NNs with a ReLU activation function and $\alpha$-Stable distributed weights. For $\alpha \in (0, 2]$, $\alpha$-Stable distributions form a broad class of heavy tails distributions, with the special case $\alpha = 2$ corresponding to the Gaussian distribution; see Samoradnitsky and Taqqu (1994) and references therein for an overview on $\alpha$-Stable distributions. According to the definition (1), we denote by $f_m(W, X, \phi; \alpha)$ the $\alpha$-Stable $\phi$-NN, namely a NN of the form (1) with the weighs $W$ distributed according to the $\alpha$-Stable distribution with $\alpha \in (0, 2)$, thus excluding the Gaussian case $\alpha = 2$. In particular, $f_m(W, X; \alpha) = f_m(W, X, \text{ReLU}; \alpha)$ denotes the $\alpha$-Stable ReLU-NN.

### 1.1.1 Related work

Neal (1996) considered $\alpha$-Stable distributions to initialize NNs' weights, showing that while all Gaussian weights vanish in the infinite-width limit, some $\alpha$-Stable weights retain a non-negligible contribution. Such a different behaviour may be attribute to the diversity of the NN's path properties as $\alpha \in (0, 2]$ varies, which makes $\alpha$-Stable NNs more flexible than Gaussian NNs; see Figure 1. Further works demonstrating practical

applications of $\alpha$-Stable NNs, with respect to Gaussian NN's, are Der and Lee (2006), Fortuin et al. (2019), Fortuin (2022), Lee et al. (2022) and Li et al. (2022); the empirical analyses developed in Fortuin et al. (2019) shows that wide $\alpha$-Stable NNs trained with gradient descent lead to a higher classification accuracy than Gaussian NNs. Motivated by these works, Favaro et al. (2020; 2021) first investigated the large with asymptotic behavior of $f_m(W, X, \phi; \alpha)$. In particular, assuming $\alpha \in (0, 2)$ and a sub-linear activation function $\phi$ it is proved that if $m \to +\infty$, then the rescaled $\alpha$-Stable $\phi$-NN $m^{-1/\alpha} f_m(W, X, \phi; \alpha)$ converges weakly to an $\alpha$-Stable process, that is a stochastic process with $\alpha$-Stable finite-dimensional distributions. See also (Jung, 2023).

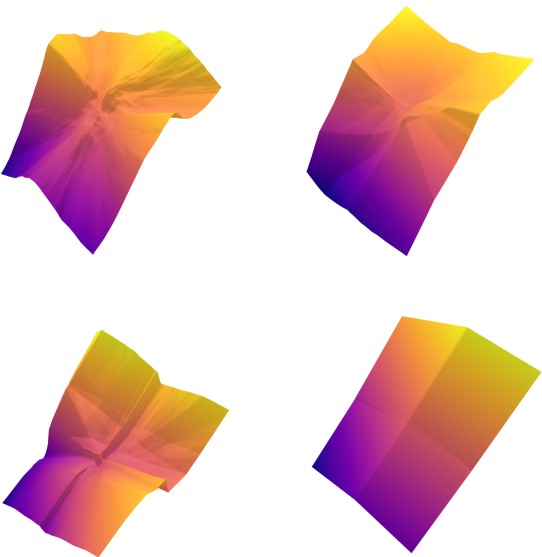

Figure 1: Sample paths of $\alpha$-Stable NNs, as a random function mapping an input in $[0, 1]^2$ to $\mathbb{R}$, with a ReLU activation function and width $m = 1024$: i) top-left panel $\alpha = 2.0$ (Gaussian distribution); ii) top-right panel $\alpha = 1.5$; iii) bottom-left panel $\alpha = 1.0$ (Cauchy distribution); iv) bottom-right panel $\alpha = 0.5$ (Lévy distribution).

### 1.1.2 Large-width asymptotics

We extend the main results of Favaro et al. (2020; 2021) to the ReLU activation function, which is arguably one of the most popular activation function in the field of NNs. In particular, we show that if $m \to +\infty$, then the rescaled $\alpha$-Stable ReLU-NN $(m \log m)^{-1/\alpha} f_m(W, X; \alpha)$ converges weakly to an $\alpha$-Stable process. For NNs with a single input, i.e. $k = 1$, the large-width limit follows by a direct application of the generalized CLT for heavy tails distributions (Uchaikin and Zolotarev, 2011; Bordino et al., 2022), whereas for $k > 1$ it requires to develop an alternative strategy that may be of independent interest in the context of multidimensional $\alpha$-Stable distributions (Samoradnitsky and Taqqu, 1994, Chapter 1 and Chapter 2). Differently from the Gaussian setting, the large-width asymptotic behaviour of $\alpha$-Stable NNs shows how the choice of the activation function $\phi$ affects the scaling of the NN. More precisely, in order to achieve the infinite-width $\alpha$-Stable process, the use of the ReLU activation in place of a sub-linear activation results in a change of the scaling $m^{-1/\alpha}$ of the NN through the additional $(\log m)^{-1/\alpha}$ term. See also Bordino et al. (2022) for a detailed discussion of this peculiar phenomenon in the context of $\alpha$-Stable ReLU-NN with a single input ($k = 1$).

### 1.1.3 Large-width training dynamics

We investigate the large-width training dynamics of $\alpha$-Stable ReLU-NNs, with the training being performed by gradient descent under the squared-error loss. In particular, consider the $\alpha$-Stable ReLU-NN $f_m(W, X; \alpha)$,

and set

$$\tilde{f}_m(W, X; \alpha) = \frac{1}{(m \log m)^{1/\alpha}} f_m(W, X; \alpha). \tag{3}$$

In analogy with (2), we define the training dynamics of $\tilde{f}_m(W(t), X; \alpha)$, with a learning rate $\eta_m$ and a $k \times k$ random matrix $H_m(W(t), X; \alpha)$ whose $(j, j')$ entry is $\langle \partial \tilde{f}_m(W(t), x_j; \alpha)/\partial W, \partial \tilde{f}_m(W(t), x_{j'}; \alpha)/\partial W \rangle$. By assuming the learning rate $\eta_m = (\log m)^{2/\alpha}$, we show that: i) if $m \to +\infty$ then $(\log m)^{2/\alpha} H_m(W(0), X; \alpha)$ converges weakly to an $(\alpha/2)$-Stable (almost surely) positive definite random matrix $\tilde{H}^*(X, X; \alpha)$; ii) and for every $\delta > 0$ the gradient descent achieves zero training error at linear rate, for $m$ sufficiently large, with probability $1 - \delta$. The limiting random matrix $\tilde{H}^*(X, X; \alpha)$ is refereed to as the $\alpha$-Stable NTK. Differently from the NTK that arises from the Gaussian setting, the $\alpha$-Stable NTK is a random kernel. More precisely, the randomness of the $\alpha$-Stable ReLU-NN at initialization does not vanish in the large-width training dynamics.

## 1.2 Organization of the paper

The paper is organized as follows. In Section 2 we characterize its large-width asymptotic behaviour of $\alpha$-Stable ReLU-NNs in terms of the infinite-width $\alpha$-Stable process. In Section 3 we characterize the large-width training dynamics of $\alpha$-Stable ReLU-NNs in terms of the $\alpha$-Stable NTK, and we show that the gradient descent achieves zero training error at linear rate, for a sufficiently large width, with high probability. Section 4 contains a discussion of our results with respect to some directions of future work. Proofs are deferred to the appendix.

## 2 Large-width asymptotics of $\alpha$-Stable ReLU-NNs

We study the large-width asymptotic behaviour of $\alpha$-Stable ReLU-NNs. The section is organized as follows: i) we recall the definition of multidimensional $\alpha$-Stable distribution (Section 2.1); ii) we define the $\alpha$-Stable ReLU-NN and characterize its large-width asymptotic behaviour in terms of the infinite-width $\alpha$-Stable process (Section 2.2); iii) we present some numerical illustrations of the large-width behaviour of $\alpha$-Stable ReLU-NNs (Section **??**). The main result of this section is Theorem 2.1, whose proof is deferred to Appendix A.1.

## 2.1 Multidimensional $\alpha$-Stable distribution

For $\alpha \in (0, 2]$, a random variable $S \in \mathbb{R}$ is distributed according to a symmetric and centered 1-dimensional $\alpha$-Stable distribution with scale $\sigma > 0$ if its characteristic function is $\mathbb{E}(\exp\{izS\}) = \exp\{-\sigma^\alpha |z|^\alpha\}$, and we write $S \sim \mathrm{St}(\alpha, \sigma)$. If the stability parameter $\alpha = 2$ then $S$ is distributed as a Gaussian distribution with mean 0 and variance $\sigma^2$. Let $\mathbb{S}^{k-1}$ be the unit sphere in $\mathbb{R}^k$, with $k \geq 1$, and let $\Gamma$ be a symmetric finite measure on $\mathbb{S}^{k-1}$. For $\alpha \in (0, 2]$, we say that a random variable $S \in \mathbb{R}^k$ is distributed according to a symmetric and centered $k$-dimensional $\alpha$-Stable distribution with spectral measure $\Gamma$ if its characteristic function is

$$\mathbb{E}(\exp\{i\langle z, S\rangle\}) = \exp\left\{-\int_{\mathbb{S}^{k-1}} |\langle z, s\rangle|^\alpha \Gamma(ds)\right\},$$

and we write $S \sim \mathrm{St}_k(\alpha, \Gamma)$. Let $1_r$ be the $r$-dimensional (column) vector with 1 in the $r$-th entry and 0 elsewhere, for any $r = 1, \ldots, k$. Then, the $r$-th element of $S$, that is $S 1_r$ is distributed as an $\alpha$-Stable distribution with scale

$$\sigma = \left(\int_{\mathbb{S}^{k-1}} |\langle 1_r, s\rangle|^\alpha \Gamma(ds)\right)^{1/\alpha}.$$

We deal mostly with $k$-dimensional $\alpha$-Stable distributions with discrete spectral measure, that is $\Gamma(\cdot) = \sum_{1 \leq i \leq n} \gamma_i \delta_{s_i}(\cdot)$ with $n \in \mathbb{N}$, $\gamma_i \in \mathbb{R}$ and $s_i \in \mathbb{S}^{k-1}$, for $i = 1, \ldots, n$ (Samoradnitsky and Taqqu, 1994, Chapter 2). All the random variables are defined on a common probability space, say $(\Omega, \mathcal{F}, \mathbb{P})$, unless otherwise stated.

We make use of the following characterization of the spectral measure of $\alpha$-stable distributions (Samoradnitsky and Taqqu, 1994, Chapter 2): if $S \sim \mathrm{St}_k(\alpha, \Gamma)$, then for every Borel set $B$ of $\mathbb{S}^{k-1}$ such that $\Gamma(\partial B) = 0$,

it holds

$$\lim_{r \to \infty} r^\alpha \mathbb{P} \left( \|S\| > r, \frac{S}{\|S\|} \in B \right) = C_\alpha \Gamma(B),$$

where

$$C_\alpha = \begin{cases} \frac{1-\alpha}{\Gamma(2-\alpha)\cos(\pi\alpha/2)} & \alpha \neq 1 \\ \frac{2}{\pi} & \alpha = 1. \end{cases} \tag{4}$$

The proof of this result is reported in Appendix B Moreover, the distribution of a random vector $\xi$ belongs to the domain of attraction of the $St_k(\alpha, \Gamma)$ distribution, with $\alpha \in (0, 2)$ and $\Gamma$ simmetric finite measure on $\mathbb{S}^{k-1}$, if and only if

$$\lim_{n \to \infty} n \mathbb{P} \left( \|\xi\| > n^{1/\alpha}, \frac{\xi}{\|\xi\|} \in A \right) = C_\alpha \Gamma(A) \tag{5}$$

for every Borel set $A$ of $S$ such that $\Gamma(\partial A) = 0$. We refer to Appendix B for more details on the derivation of (5). See also Samoradnitsky and Taqqu (1994, Chapter 1 and Chapter 2) for further details on the constant $C_\alpha$.

## 2.2 The infinite-width $\alpha$-Stable process

To define a generic ReLU NN, let us consider the following elements: i) for $d, k \geq 1$ let $X$ be the $d \times k$ NN's input, such that $x_j = (x_{j1}, \ldots, x_{jd})^T$ is the $j$-th input (column vector); ii) for $m \geq 1$ let $W = (w_1^{(0)}, \ldots, w_m^{(0)}, w)$ be the NN's weights, such that $w_i^{(0)} = (w_{i1}^{(0)}, \ldots, w_{id}^{(0)})$ and $w = (w_1, \ldots, w_m)$. A ReLU-NN of width $m$ is

$$f_m(W, X; \alpha) = (f_m(W, x_1; \alpha), \ldots, f_m(W, x_k; \alpha)), \tag{6}$$

where

$$f_m(W, x_j; \alpha) = \sum_{i=1}^m w_i \langle w_i^{(0)}, x_j \rangle I(\langle w_i^{(0)}, x_j \rangle > 0) \qquad j = 1, \ldots, k,$$

with $I(\cdot)$ being the indicator function. We denote by $W(0) = (w_1^{(0)}(0), \ldots, w_m^{(0)}(0), w(0))$ the NN's weights at random initialization. If the NN's weights $w_{ij}^{(0)}$'s and the $w_i$'s are initialized as i.i.d. $\alpha$-Stable random variables, with $\alpha \in (0, 2)$ and $\sigma > 0$, then $f_m(W(0), X; \alpha)$ defines an $\alpha$-Stable ReLU-NN of width $m$.

**Theorem 2.1.** *For any $\alpha \in (0, 2)$, let $f_m(W(0), X; \alpha)$ be an $\alpha$-Stable ReLU-NN of width $m$. If $m \to +\infty$ then*

$$\frac{1}{(m \log m)^{1/\alpha}} f_m(W(0), X; \alpha) \xrightarrow{w} f(X),$$

*where $f(X) \sim St_k(\alpha, \Gamma_X)$, with*

$$\Gamma_X = \frac{C_\alpha}{4} \sum_{i=1}^d (\|[x_{ji}I(x_{ji} > 0)]_j\|^\alpha) D_i^+(X) + \|[x_{ji}I(x_{ji} < 0)]_j\|^\alpha) D_i^-(X)$$

*such that*

$$D_i^+(X) = \delta \left( \frac{[x_{ji}I(x_{ji} > 0)]_j}{\|[x_{ji}I(x_{ji} > 0)]_j\|} \right) + \delta \left( -\frac{[x_{ji}I(x_{ji} > 0)]_j}{\|[x_{ji}I(x_{ji} > 0)]_j\|} \right)$$

*and*

$$D_i^-(X) = \delta \left( \frac{[x_{ji}I(x_{ji} < 0)]_j}{\|[x_{ji}I(x_{ji} < 0)]_j\|} \right) + \delta \left( -\frac{[x_{ji}I(x_{ji} < 0)]_j}{\|[x_{ji}I(x_{ji} < 0)]_j\|} \right),$$

*where, for any $s \in \mathbb{S}^{k-1}$, $\delta(s)$ is the probability measure degenerate in $s$, and $C_\alpha$ is in (4). The stochastic process $f(X) = (f(x_1), \ldots, f(x_k))$, as a process indexed by $X$, is an $\alpha$-Stable process with spectral measure $\Gamma_X$.*

*Sketch of the proof of Theorem 2.1.* The $\alpha$-stable ReLU-NN of width $m$ is a sum of $m$ independent and identically distributed random vectors. The proof relies on the analysis of the tail behavior of these summands, and it exploits a characterization of the multivariate $\alpha$-Stable distribution as the limiting distribution

of sums of independent random vectors that exhibit specific tail properties. We refer to Appendix A.1 for details.

For a broad class of bounded or sub-linear activation functions, Favaro et al. (2021) characterizes the large-width distribution of deep $\alpha$-Stable NNs. See also Bordino et al. (2022) and references therein. In particular, let

$$f_m(W, x_j, \phi; \alpha) = \sum_{i=1}^{m} w_i \phi\langle w_i^{(0)}, x_j\rangle$$

be the $\alpha$-Stable $\phi$-NN of width $m$ for the input $x_j$, for $j = 1, \ldots, k$, with $\phi$ being a bounded activation function. Let $f_m(X; \alpha) = (f_m(x_1; \alpha), \ldots, f_m(x_k; \alpha))$. From Favaro et al. (2021, Theorem 1.2), if $m \to +\infty$ then

$$\frac{1}{m^{1/\alpha}} f_m(W, X, \phi; \alpha) \xrightarrow{\text{w}} f(X), \tag{7}$$

with $f(X)$ being an $\alpha$-Stable process with spectral measure $\Gamma_{X,\phi}$. Theorem 2.1 extends Favaro et al. (2021, Theorem 1.2) to the ReLU activation function. Theorem 2.1 shows that the use of the ReLU activation in place of a bounded activation results in a change of the scaling $m^{-1/\alpha}$ in (7), through the inclusion of the $(\log m)^{-1/\alpha}$ term. This is a critical difference between the $\alpha$-Stable setting and Gaussian setting, as in the latter the choice of the activation function $\phi$ does not affect the scaling $m^{-1/2}$ required to achieve the infinite-width Gaussian process. For $k = 1$, we refer to Bordino et al. (2022) for a detailed analysis of infinitely wide limits of $\alpha$-Stable NNs with general classes of sub-linear, linear and super-linear activation functions.

**Remark 2.1.** *The need of the additional* $\log(m)$ *can be clarified by considering the $\alpha$-Stable ReLU-NN with a single input, i.e. $k = 1$, where the proof of Theorem 2.1 reduces to a straightforward application of the generalized CLT for heavy tails distributions (Uchaikin and Zolotarev, 2011; Bordino et al., 2022). In particular, we refer to Theorem 2.1. and Theorem 2.6 of Bordino et al. (2022), which show how the* $\log$ *term arises from the tail behaviour of the product of $\alpha$-Stable random variable $w_i w_i^{(0)}$ 's, which defines the NN; see Cline (1986) and references therein. The* $\log$ *term is expected to hold for any activation that has a linear growth.*

To demonstrate numerically Theorem 2.1, we sample random neural networks according to 3 for various values of width $m$ and stability index $\alpha$. We evaluate these networks on a fine uniform grid of points in $[0, 1]^2$. Figure 2 displays the results, which show that the function samples remain well-behaved as $m$ grows larger.

## 3 Large-width training dynamics of $\alpha$-Stable ReLU-NNs

We study the large-width training dynamics of $\alpha$-Stable ReLU-NNs. The section is organized as follows: we define the training dynamics of the $\alpha$-Stable ReLU-NN and characterize its large-width asymptotic behaviour in terms of the $\alpha$-Stable NTK (Section 3.1); ii) we show that the gradient descent achieves zero training error at linear rate, for a sufficiently large width, with high probability (Section 3.2). The main results of this section are Theorem 3.1 and Theorem 3.2, whose proofs are deferred to Appendix A.2 and Appendix A.4, respectively.

### 3.1 The $\alpha$-Stable NTK

Let $f_m(W, X; \alpha)$ be the $\alpha$-Stable ReLU-NN defined in (6), with $\alpha \in (0, 2)$, and let $(X, Y)$ be the training set, where $Y = (y_1, \ldots, y_k)$ is the (training) output such that $y_j$ corresponds to the $j$-th input $x_j$. Then, we set

$$\tilde{f}_m(W, X; \alpha) = \frac{1}{(m \log m)^{1/\alpha}} f_m(W, X; \alpha),$$

such that $\tilde{f}_m(W, x_j; \alpha) = (m \log m)^{-1/\alpha} f_m(W, x_j; \alpha)$ is the (model) output of the $j$-th input $x_j$, for $j = 1, \ldots, k$. Assuming the squared-error loss function $\ell(y_j, \tilde{f}_m(W, x_j; \alpha)) = 2^{-1} \sum_{1 \le j \le k} (\tilde{f}_m(W, x_j; \alpha) - y_j)^2$, by

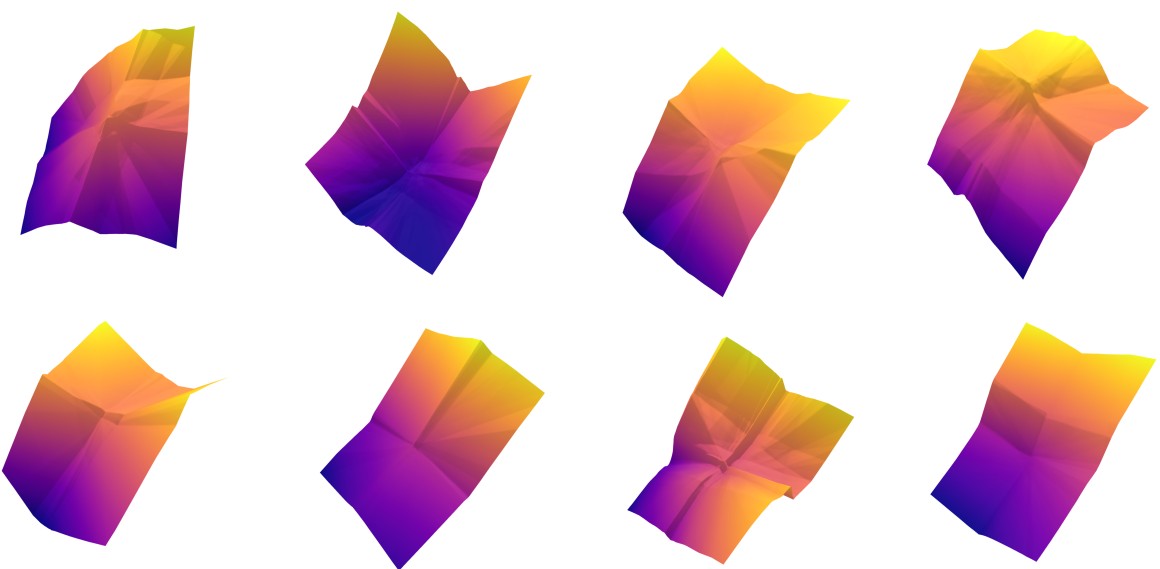

Figure 2: Sample paths of $\alpha$-Stable NNs, as a random function mapping an input in $[0,1]^2$ to $\mathbb{R}$, with a ReLU activation function; width values (left to right): $m = 64, 256, 1024, 4096$; $\alpha = 1.5$ (top panel), $\alpha = 1.0$ (bottom panel).

a direct application of the chain rule we obtain the training dynamics of $\tilde{f}_m(W, X; \alpha)$. In particular, for any $t \geq 0$

$$\frac{\mathrm{d}\tilde{f}_m(W(t), X; \alpha)}{\mathrm{d}t} = -(\tilde{f}_m(W(t), X; \alpha) - Y)\eta_m H_m(W(t), X), \tag{8}$$

where the kernel $H_m(W(t), X)$ in the NN's training dynamics is a $k \times k$ random matrix whose $(j, j')$ entry is

$$H_m(W(t), X)[j, j'] = \left\langle \frac{\partial \tilde{f}_m(W(t), x_j; \alpha)}{\partial W}, \frac{\partial \tilde{f}_m(W(t), x_{j'}; \alpha)}{\partial W} \right\rangle, \tag{9}$$

and $\eta_m$ is the learning rate. The training dynamics for $\tilde{f}_m(W, X; \alpha)$ is standard, and if follows training dynamics presented in Section 1 for the Gaussian setting. See (Arora et al., 2019, Section 3) and references therein for details.

For the training dynamics (8), we study the large-width behaviour of $H_m(W(0), X)$ in (9). In particular, we set

$$\tilde{H}_m(W(0), X) = (\log m)^{2/\alpha} H_m(W(0), X), \tag{10}$$

and show that if $m \rightarrow +\infty$, then $\tilde{H}_m(W(0), X)$ converges weakly to a positive definite random matrix $\tilde{H}^*(X, X, \alpha)$ distributed according to an $(\alpha/2)$-Stable distribution. To prove this result it useful to decompose $\tilde{H}_m(W(0), X)$ as

$$\tilde{H}_m(W(0), X) = \tilde{H}_m^{(1)}(W(0), X) + \tilde{H}_m^{(2)}(W(0), X), \tag{11}$$

where

$$\tilde{H}_m^{(1)}(W(0), X)[j, j'] = \frac{1}{m^{2/\alpha}} \sum_{i=1}^m w_i^2 \langle x_j, x_{j'} \rangle I(\langle w_i^{(0)}, x_j \rangle > 0) I(\langle w_i^{(0)}, x_{j'} \rangle > 0), \tag{12}$$

and

$$\tilde{H}_m^{(2)}(W(0), X)[j, j'] = \frac{1}{m^{2/\alpha}} \sum_{i=1}^m \langle w_i^{(0)}, x_j \rangle I(\langle w_i^{(0)}, x_j \rangle > 0) \langle w_i^{(0)}, x_{j'} \rangle I(\langle w_i^{(0)}, x_{j'} \rangle > 0), \tag{13}$$

respectively. The next theorem characterizes the large-width asymptotic behaviour of $\tilde{H}_m(W(0), X)$ in terms of the $\alpha$-Stable NTK.

**Theorem 3.1.** *For any $\alpha \in (0,2)$, let $\tilde{H}_m(W(0), X)$, $\tilde{H}_m^{(1)}(W(0), X)$ and $\tilde{H}_m^{(2)}(W(0), X)$ be the $k \times k$ random matrices whose $(j, j')$ entries are defined in (10), (12), and (13), respectively. Moreover, for every $k \geq 1$ and $u \in \{0,1\}^k$, let*

$$B_u = \{v \in \mathbb{R}^d : \langle v, x_j \rangle > 0 \text{ if } u_j = 1, \langle v, x_j \rangle \leq 0 \text{ if } u_j = 0, j = 1, \ldots, k\},$$

*and for every $i = 1, \ldots, d$, let $e_i$ be a $d$-dimensional vector such that $e_{ij} = 1$ for $j = i$ and $e_{ij} = 0$ for $j \neq i$. As $m \to +\infty$,*

$$(\tilde{H}_m^{(1)}(W(0), X), \tilde{H}_m^{(2)}(W(0), X)) \overset{w}{\longrightarrow} (\tilde{H}_1^*(\alpha), \tilde{H}_2^*(\alpha)),$$

*where $\tilde{H}_1^*(\alpha)$ and $\tilde{H}_2^*(\alpha)$ are $k \times k$ random matrices that are stochastically independent, positive semi-definite, and distributed according to $(\alpha/2)$-Stable distributions with spectral measures $\Gamma_1^*$ and $\Gamma_2^*$, respectively, such that*

$$\Gamma_1^* = C_{\alpha/2} \sum_{u \in \{0,1\}^k} \mathbb{P}(w_i^{(0)}(0) \in B_u) \frac{\delta\left(\frac{[\langle x_j, x_{j'}\rangle u_j u_{j'}]_{j,j'}}{(\sum_{j,j'} \langle x_j, x_{j'}\rangle^2 u_j u_{j'})^{1/2}}\right)}{\left(\sum_{j,j'} \langle x_j, x_{j'}\rangle^2 u_j u_{j'}\right)^{-\alpha/4}}, \tag{14}$$

*and*

$$\Gamma_2^* = C_{\alpha/2} \sum_{u \in \{0,1\}^k} \sum_{\{i : \{e_i, -e_i\} \cap B_u \neq \emptyset\}} \frac{\delta\left(\frac{[x_{ji} u_j x_{j'i} u_{j'}]_{j,j'}}{\sum_j x_{ji}^2 u_j}\right)}{\left(\sum_j x_{ji}^2 u_j\right)^{-\alpha/2}}, \tag{15}$$

*with $C_{\alpha/2}$ in (4). Furthermore, as $m \to \infty$,*

$$\tilde{H}_m(W(0), X) \overset{w}{\longrightarrow} \tilde{H}^*(X, X; \alpha),$$

*where $\tilde{H}^*(X, X; \alpha)$ is a $k \times k$ random matrix that is positive semi-definite and distributed according to an $(\alpha/2)$-Stable distribution with spectral measure $\Gamma^* = \Gamma_1^* + \Gamma_2^*$. $\tilde{H}^*(X, X; \alpha)$ is refereed to as the $\alpha$-Stable NTK.*

*Sketch of the proof of Theorem 3.1.* We can see $(\tilde{H}_m^{(1)}(W(0), X), \tilde{H}_m^{(2)}(W(0), X))$ as a random vector of dimension $2k^2$, with $k \geq 1$, whose elements are sums of independent and identically distributed random vectors. The proof relies on the analysis of the tail behavior of these summands, and it exploits a characterization of the multivariate $\alpha$-Stable distribution as limiting distribution of the sum of independent and identically distributed random vectors that exhibit specific tail properties. We refer to Appendix A.2 for the details.

It turns out that the $(\alpha/2)$-Stable distributions of the limiting random matrices $\tilde{H}_1^*(\alpha)$ and $\tilde{H}_2^*(\alpha)$ are absolutely continuous in suitable subspaces of the space of symmetric and positive semi-definite matrices; see Lemma A.4 and Lemma A.5 for details on the distribution of the random matrix $\tilde{H}_1^*(\alpha)$, and Lemma A.6 and Lemma A.7 for details on the distribution of the random matrix $\tilde{H}_2^*(\alpha)$. This is applied in the next theorem to show that the minimum eigenvalues of $\tilde{H}_m^{(1)}(W(0), X)$ and of $\tilde{H}_m^{(2)}(W(0), X)$ are bounded away from zero, uniformly in $m$, for $m$ sufficiently large, with arbitrarily high probability. Accordingly, the minimum eigenvalue of $\tilde{H}_m(W(0), X) = \tilde{H}_m^{(1)}(W(0), X) + H_m^{(2)}(W(0), X)$ is bounded away from zero, uniformly in $m$, for $m$ sufficiently large, with arbitrarily high probability. We denote by $\lambda_{\min}(\cdot)$ the minimum eigenvalue.

**Proposition 3.1.** *For any $\alpha \in (0,2)$, let $\tilde{H}_m(W, X)$, $\tilde{H}_m^{(1)}(W, X)$ and $\tilde{H}_m^{(2)}(W, X)$ be the random matrices as in Theorem 3.1. For every $\delta > 0$ there exist strictly positive numbers $\lambda_0$, $\lambda_1$ and $\lambda_2$ such that, for $m$ sufficiently large,*

$$\lambda_{min}(\tilde{H}_m^{(i)}(W(0), X)) > \lambda_i \qquad i = 1, 2,$$

*and*

$$\lambda_{min}(\tilde{H}_m(W(0), X)) > \lambda_0.$$

*with probability at least $1 - \delta$.*

See Appendix A.3 for the proof of Proposition 3.1. Theorem 3.1 and Proposition 3.1 provide an extension of some of the main results of Jacot et al. (2018) to the setting of $\alpha$-Stable ReLU NN, for $\alpha \in (0,2)$. See also Du et al. (2019), Arora et al. (2019), Lee et al. (2019) and references therein. In particular, our results show that

i) as $m \to +\infty$, the random matrix $(\log m)^{2/\alpha} H_m(W(0), X)$ converges weakly to the $\alpha$-Stable NTK $\tilde{H}^*(X, X; \alpha)$, such that $\tilde{H}^*(X, X; \alpha)$ is a $(\alpha/2)$-Stable (almost surely) positive definite random matrix;

ii) at random initialization for the $\alpha$-Stable ReLU-NN, for every $\delta > 0$ the minimum eigenvalue of the random matrix $\tilde{H}_m(W(0), X)$ remains bounded away from zero, for $m$ sufficiently large, with probability $1 - \delta$.

Differently from the NTK that arises from the Gaussian setting, the $\alpha$-Stable NTK is a random kernel. That is, the randomness of the $\alpha$-Stable ReLU-NN at initialization does not vanish in the large-width training dynamics. Such a randomness makes more challenging the study of the corresponding large-with training dynamics.

### 3.2 Zero training error at linear rate

Under the training dynamics (8), we show that for every $\delta > 0$ the gradient descent achieves zero training error at linear rate, for $m$ sufficiently large, with probability $1 - \delta$. In order to prove this result we combine Proposition 3.1 with the next proposition, which shows that, if $m$ is sufficiently large, then with high probability the minumum eigenvalue of the random matrix $\tilde{H}_m(W(t), X)$ remains bounded away from zero. We denote by $\| \cdot \|_F$ and $\| \cdot \|_2$ the Frobenius and operator norms of symmetric and positive semi-definite matrices, respectively.

**Proposition 3.2.** *Let $\gamma \in (0,1)$ and $c > 0$. For $k \geq 1$ let the NN's inputs $x_1, \ldots, x_k$ be linearly independent and such that $\|x_j\| = 1$. For any $\alpha \in (0,2)$, let $\tilde{H}_m(W, X)$ and $\tilde{H}_m^{(2)}(W, X)$ be the random matrices as in Theorem 3.1. For every $\delta > 0$ the following properties hold for every $t \geq 0$, with probability at least $1 - \delta$, for $m$ sufficiently large:*

*(i) for every $j = 1, \ldots, k$,*

$$(\log m)^{2/\alpha} \left\| \frac{\partial \tilde{f}_m}{\partial w}(W(t), x_j; \alpha) - \frac{\partial \tilde{f}_m}{\partial w}(W(0), x_j; \alpha) \right\|_F^2 < cm^{-2\gamma/\alpha};$$

*(ii) there exists $\lambda_0 > 0$ such that*

$$\|\tilde{H}_m^{(2)}(W(t), X) - \tilde{H}_m^{(2)}(W(0), X)\|_F < \lambda_0 m^{-\gamma/\alpha}$$

*and*

$$\lambda_{min}(\tilde{H}_m(W(t), X)) > \frac{\lambda_0}{2}.$$

*Sketch of the proof of Proposition 3.2.* The inequality displayed in (i) holds as long as $W(t)$ stays within a neighborhood of $W(0)$ with radius on the order of $(\log m)^{2/\alpha}$, and viceversa. This implies that the fluctuations of $\partial \tilde{f}(W(t), X)/\partial w$ during the training of the $\alpha$-Stable ReLU-NN vanish as $m \to \infty$. Consequently, the first inequality displayed in (ii) also holds throughout training if $m$ is large enough. Together with Proposition 3.1, this ensures that the minimum eigenvalue of the random matrix $\tilde{H}_m^{(2)}(W(t), X)$ remains bounded away from zero during training. The same argument applies to the random matrix $\tilde{H}_m(W(t), X)$, which is the sum of $\tilde{H}_m^{(2)}(W(t), X)$ and of the non-negative definite matrix $\tilde{H}_m^{(1)}(W(t), X)$. We refer to Appendix A.4 for the details.

Now, we are in the position to show that the gradient descent achieves zero training error at linear rate, for a sufficiently large width, with high probability. From Proposition 3.2, for a fixed $\delta > 0$, let $m$ and $\lambda_0 > 0$ be such that

$$\lambda_{\min}(\tilde{H}_m(W(s), X)) > \frac{\lambda_0}{2}.$$

for every $s \leq t$, on a set $N \in \mathcal{F}$ with $\mathbb{P}[N] > 1 - \delta$. Accordingly, for any random initialization $W(0)(\omega)$, with $\omega \in N$,

$$\frac{\mathrm{d}}{\mathrm{d}s}\|Y - \tilde{f}_m(W(s)(\omega), X; \alpha)\|_2^2 \leq -\lambda_0\|Y - \tilde{f}_m(W(s)(\omega), X; \alpha)\|_2^2,$$

and hence

$$\frac{\mathrm{d}}{\mathrm{d}s}\exp(\lambda_0 s)\|Y - \tilde{f}_m(W(s)(\omega), X; \alpha)\|_2^2 \leq 0.$$

Therefore, by observing that $\exp(\lambda_0 s)\|Y - \tilde{f}_m(W(s)(\omega), X; \alpha)\|_2^2$ is a decreasing function of $s > 0$, then we write

$$\|Y - \tilde{f}_m(W(s)(\omega), X; \alpha)\|_2^2 \leq \exp(-\lambda_0 s)\|Y - \tilde{f}_m(W(0)(\omega), X; \alpha)\|_2^2.$$

In the next theorem we summarize the main finding on the large-width training dynamics of $\alpha$-Stable ReLU NNs.

**Theorem 3.2.** *For $k \geq 1$ let the NN's inputs $x_1, \ldots, x_k$ be linearly independent and such that $\|x_j\| = 1$. For any $\alpha \in (0, 2)$, under the training dynamics (8), if the learning rate $\eta_m = (\log m)^{2/\alpha}$ then for every $\delta > 0$ there exists $\lambda_0 > 0$ such that, for $m$ sufficiently large and any $t > 0$, with probability at least $1 - \delta$ it holds true that*

$$\|Y - \tilde{f}_m(W(t), X; \alpha)\|_2^2 \leq \exp(-\lambda_0 t)\|Y - \tilde{f}_m(W(0), X; \alpha)\|_2^2.$$

## 4 Discussion

In this paper, we investigated large-width asymptotics and training dynamics of $\alpha$-Stable ReLU-NNs, namely NNs with a ReLU activation function and $\alpha$-Stable distributed weights. With regards to the large-width asymptotics, our result (Theorem 2.1) extends the main result of Favaro et al. (2020; 2021) to the ReLU activation function, showing the need of an additional logarithmic term in the scaling of the NN to achieve the infinite-width $\alpha$-Stable process. With regards to the large-width training dynamics, our results (Theorem 3.1 and Theorem 3.2) extends some of the main results of Jacot et al. (2018) to $\alpha$-Stable ReLU-NNs, showing that randomness of the $\alpha$-Stable ReLU-NN at initialization does not vanish in the large-width training dynamics.

It remains open to establish a large-width equivalence between training an $\alpha$-Stable ReLU-NN and performing a kernel regression with the $\alpha$-Stable NTK. For Gaussian NN, Jacot et al. (2018) showed that during training $t > 0$, if $m$ is sufficiently large then the fluctuations of the squared Frobenious norm $\|H_m(W(t), X) - H_m(W(0), X)\|_F^2$ are vanishing. This suggested to replace $\eta_m H_m(W(t), X)$ with the NTK $H^*(X, X)$ in the dynamics (2), and write

$$\frac{\mathrm{d}f^*(t, X)}{\mathrm{d}t} = -(f^*(t, X) - Y)H^*(X, X).$$

This is the dynamics of a kernel regression under gradient flow, for which at $t \to +\infty$ the prediction for a generic test point $x \in \mathbb{R}^d$ is of the form $f^*(x) = YH^*(X, X)^{-1}H^*(X, x)^T$. In particular, the prediction of the Gaussian NN $\tilde{f}_m(W(t), x)$ at $t \to +\infty$, for $m$ sufficiently large, is equivalent to the kernel regression prediction $f^*(x)$ (Arora et al., 2019). Within the $\alpha$-Stable setting, it is not clear whether the fluctuations of $\tilde{H}_m(W(t), X) = \tilde{H}_m^{(1)}(W(t), X) + \tilde{H}_m^{(2)}(W(t), X)$ during the training vanish, as $m \to \infty$. Proposition 3.2 shows that the fluctuations of $\tilde{H}_m^{(2)}(W(t), X)$ vanish, as $m \to \infty$. Such a result is based on the fact that for every $\delta > 0$ it holds that

$$(\log m)^{2/\alpha}\left\|\frac{\partial \tilde{f}_m}{\partial w}(W, x_j; \alpha) - \frac{\partial \tilde{f}_m}{\partial w}(W(0), x_j; \alpha)\right\|_F^2 < cm^{-2\gamma/\alpha},$$

for every $j = 1, \ldots, k$, and for every $W$ such that $\|W - W(0)\|_F \leq (\log m)^{2/\alpha}$, with probability at least $1 - \delta$, if $m$ is sufficiently large. In particular, we refer to Lemma A.8 for details. The same large-width property is not true if the partial derivatives with respect to $w$ are replaced by the partial derivatives with respect to $w^{(0)}$. Accordingly, it is not clear whether the fluctuations of $\tilde{H}_m^{(1)}(W(t), X)$ during training also vanish, as $m \to \infty$.

Another interesting avenue for future research would be to extend our results to the more general setting of deep NNs, with $D \geq 2$ being the depth. Let us consider the following setting: i) for $d, k \geq 1$ let $X$ be the $d \times k$ NN's input, with $x_j = (x_{j1}, \ldots, x_{jd})^T$ being the $j$-th input (column vector); ii) for $D, m \geq 1$ and $n \geq 1$ let: i) $(W^{(1)}, \ldots, W^{(D)})$ be the NN's weights such that $W^{(1)} = (w_{1,1}^{(1)}, \ldots, w_{m,d}^{(1)})$ and $W^{(l)} = (w_{1,1}^{(l)}, \ldots, w_{m,m}^{(1)})$ for $2 \leq l \leq D$, where the $w_{i,j}^{(l)}$'s are i.i.d. as an $\alpha$-Stable distribution with scale $\sigma > 0$, e.g. we can assume $\sigma = 1$. Then,

$$f_i^{(1)}(X; \alpha) = \sum_{j=1}^{d} w_{i,j}^{(1)} x_j$$

and

$$f_{i,m}^{(l)}(X; \alpha) = \sum_{j=1}^{m} w_{i,j}^{(l)} f_j^{(l-1)}(X, m) I(f_j^{(l-1)}(X, m) > 0)$$

with $f_{i,m}^{(1)}(X; \alpha) = f_i^{(1)}(X; \alpha)$, is a deep $\alpha$-Stable ReLU-NN of depth $D$ and width $m$. If the NN's width grows sequentially over the NN's layers, i.e. $m \to +\infty$ one layer at a time, it is easy to extend Theorem 2.1 to $f_{i,m}^{(l)}(X; \alpha)$. Under the same assumption on the growth of $m$, we expect the analysis of the large-width training dynamics to follow along lines similar to that of Theorem 3.1 and Theorem 3.2, though computations may be more involved. A more challenging task would to extend our results to deep $\alpha$-Stable ReLU-NNs under the assumptions that the NN's width grows jointly over the NN's layers, i.e. $m \to +\infty$ simultaneously over the layers

## Acknowledgments

The authors wish to thank the Action Editor, Professor Murat A. Erdogdu, and three anonymous Referees for their helpful suggestions. Stefano Favaro was funded by the European Research Council under the Horizon 2020 research and innovation programme, grant 817257. Stefano Favaro also gratefully acknowledge support from the Italian Ministry of Education, University and Research, "Dipartimenti di Eccellenza" grant 2023-2027.

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

# A

## A.1 Proof of Theorem 2.1

To simplify the notation, we set in this section: $w := w(0)$, $w^{(0)} := w^{(0)}(0)$, and $W := W(0)$. First, we will prove that $[\langle w_i^{(0)}, x_j \rangle I(\langle w_i^{(0)}, x_j \rangle > 0)]_j$ belongs to the domain of attraction of an $\alpha$-stable law with spectral measure

$$\Gamma_1 = C_\alpha \mathbb{E}_{u \sim \Gamma_0} \left( \|[\langle u, x_j \rangle I(\langle u, x_j \rangle > 0)]_j\|^\alpha \delta \left( \frac{[\langle u, x_j \rangle I(\langle u, x_j \rangle > 0)]_j}{\|[\langle u, x_j \rangle I(\langle u, x_j \rangle > 0)]_j\|} \right) \right),$$

where $\Gamma_0$ is the spectral measure of $w_i^{(0)}$. For this, it is sufficient to show that

$$r^\alpha \mathbb{P} \left( \frac{[\langle w_i^{(0)}, x_j \rangle I(\langle w_i^{(0)}, x_j \rangle > 0)]_j}{\|[\langle w_i^{(0)}, x_j \rangle I(\langle w_i^{(0)}, x_j \rangle > 0)]_j\|} \in B, \|[\langle w_i^{(0)}, x_j \rangle I(\langle w_i^{(0)}, x_j \rangle > 0)]_j\| > r \right)$$
$$\to C_\alpha \Gamma_1(B),$$

for every Borel set $B$ of $\mathbb{S}^{k-1}$ such that $\Gamma_1(\partial B) = 0$ (see Appendix B). Let $T : \mathbb{S}^{k-1} \mapsto [0,1]^k$ and $C : \mathbb{R}^k \setminus \{0\} \to \mathbb{S}^{k-1}$ be defined as $T(u) = [\langle u, x_j \rangle I(\langle u, x_j \rangle > 0]_j$ and $C(v) = v/\|v\|$, respectively. Fix a Borel set $B$ of $\mathbb{S}^{k-1}$ such that $\Gamma_1(\partial B) = 0$. This condition implies that

$$\Gamma_0 \left( \{ u \in \mathbb{S}^{k-1} : \|T(u)\| \neq 0, T(u) \in C^{-1}(\partial B) \} \right)$$
$$= \Gamma_0 \left( \left\{ u \in \mathbb{S}^{k-1} : \|T(u)\| \neq 0, \frac{T(u)}{\|T(u)\|} \in \partial B \right\} \right) = 0.$$

Hence

$$\Gamma_0 \left( T^{-1} \left( \{ z \in [0,1]^k : \|z\| \neq 0, z \in \partial C^{-1}(B) \} \right) \right)$$
$$= \Gamma_0 \left( T^{-1} \left( \{ z \in [0,1]^k : \|z\| \neq 0, z \in C^{-1}(\partial B) \} \right) \right) = 0.$$

Now, let $Z = T(w_i^{(0)}/\|w_i^{(0)}\|)I(\|w_i^{(0)}\| \neq 0)$. We can write that

$$r^\alpha \mathbb{P} \left( \frac{[\langle w_i^{(0)}, x_j \rangle I(\langle w_i^{(0)}, x_j \rangle > 0)]_j}{\|[\langle w_i^{(0)}, x_j \rangle I(\langle w_i^{(0)}, x_j \rangle > 0)]_j\|} \in B, \|[\langle w_i^{(0)}, x_j \rangle I(\langle w_i^{(0)}, x_j \rangle > 0)]_j\| > r \right)$$
$$= r^\alpha \mathbb{P} \left( \|Z\| \neq 0, \frac{Z}{\|Z\|} \in B, \|w_i^{(0)}\|\|Z\| > r \right)$$
$$= \int_{C^{-1}(B) \cap [0,1]^k} r^\alpha \mathbb{P}(\|w_i^{(0)}\| > r\|z\|^{-1}, Z \in dz)$$
$$= \int_{C^{-1}(B) \cap [0,1]^k} \|z\|^\alpha (r\|z\|^{-1})^\alpha \mathbb{P}(\|w_i^{(0)}\| > r\|z\|^{-1}, \frac{w_i^{(0)}}{\|w_i^{(0)}\|} \in T^{-1}(dz)).$$

Since $\Gamma_0 \left( T^{-1} \left( \{ z \in [0,1]^k : z \neq 0, z \in \partial(C^{-1}(B)) \} \right) \right) = 0$, then the points of discontinuity of the function $\|z\|^\alpha I(C^{-1}(B))(z)$ have zero $\Gamma_0(T^{-1}(\cdot))$-measure. It follows that

$$\int_{C^{-1}(B) \cap [0,1]^k} \|z\|^\alpha (r\|z\|^{-1})^\alpha \mathbb{P}(\|w_i^{(0)}\| > r\|z\|^{-1}, w_i^{(0)} \in T^{-1}(dz))$$
$$\to C_\alpha \int_{C^{-1}(B) \cap [0,1]^k} \|z\|^\alpha \Gamma_0(T^{-1}(dz))$$
$$= C_\alpha \int_{\mathbb{S}^{k-1}} I(u \in B) \left( \frac{T(u)}{\|T(u)\|} \right) \|T(u)\|^\alpha \Gamma_0(du)$$
$$= C_\alpha \Gamma_1(B),$$

as $r \to \infty$, which completes the proof that $[\langle w_i^{(0)}, x_j \rangle I(\langle w_i^{(0)}, x_j \rangle > 0)]_j$ belongs to the domain of attraction of an $\alpha$-stable law with spectral measure $\Gamma_1$. Then, for every $k$-dimensional vector $s$,

$$\frac{1}{m^{1/\alpha}} \sum_{i=1}^{m} \sum_{j=1}^{k} s_j \langle w_i^{(0)}, x_j \rangle I(\langle w_i^{(0)}, x_j \rangle > 0),$$

as a sequence of random variables in $m$, converges in distribution, as $m \to +\infty$, to a random variable with $\alpha$-stable distribution and characteristic function

$$\exp \left( -|t|^\alpha \mathbb{E}_{u \sim \Gamma_0} \left( |\sum_{j=1}^{k} s_j \langle u, x_j \rangle I(\langle u, x_j \rangle > 0)|^\alpha \right) \right).$$

Thus, the distribution of $\sum_{j=1}^{k} s_j \langle w_i^{(0)}, x_j \rangle I(\langle w_i^{(0)}, x_j \rangle > 0)$ belongs to the domain of attraction of an $\alpha$-stable law. In particular, this implies that as $m \to +\infty$

$$r^\alpha \mathbb{P} \left( |\sum_{j=1}^{k} s_j \langle w_i^{(0)}, x_j \rangle I(\langle w_i^{(0)}, x_j \rangle > 0)| > r \right)$$

$$\to C_\alpha \mathbb{E}_{u \sim \Gamma_0} \left( |\sum_{j=1}^{k} s_j \langle u, x_j \rangle I(\langle u, x_j \rangle > 0)|^\alpha \right).$$

We now study the tail behaviour of $\left| w_i \sum_{j=1}^{k} s_j \langle w_i^{(0)}, x_j \rangle I(\langle w_i^{(0)}, x_j \rangle > 0) \right|$. By Cline (1986, Section 5),

$$\mathbb{P} \left( |w_i| \ |\sum_{j=1}^{k} s_j \langle w_i^{(0)}, x_j \rangle I(\langle w_i^{(0)}, x_j \rangle > 0)| > e^t \right) = \overline{F * G}(t),$$

where

$$\overline{F}(t) = \mathbb{P} \left( |w_i| > e^t \right), \qquad \overline{G}(t) = \mathbb{P} \left( |\sum_{j=1}^{k} s_j \langle w_i^{(0)}, x_j \rangle I(\langle w_i^{(0)}, x_j \rangle > 0)| > e^t \right).$$

We now prove that $F$ and $G$ satisfy the assumptions of Cline (1986, Theorem 4) with $\beta = \gamma = 0$. The distribution functions $F$ and $G$ have exponential tails with rate $\alpha$. Indeed, for all real $u$,

$$\lim_{t \to \infty} \frac{\overline{F}(t - u)}{\overline{F}(t)} = \lim_{t \to \infty} \frac{\mathbb{P}(|w_i| > e^{t-u})}{\mathbb{P}(|w_i| > e^t)} = \frac{e^{-\alpha(t-u)}}{e^{-\alpha t}} = e^{\alpha u}.$$

Analogously for $G$. Moreover the functions $b(t) = e^{\alpha t} \overline{F}(t)$ and $c(t) = e^{\alpha t} \overline{G}(t)$ are regularly varying with exponent zero: for all $y > 0$,

$$\lim_{t \to \infty} \frac{b(yt)}{b(t)} = \lim_{t \to \infty} \frac{e^{\alpha yt} \mathbb{P}(|w_i| > e^{yt})}{e^{\alpha t} \mathbb{P}(|w_i| > e^t)} = \lim_{t \to \infty} \frac{e^{\alpha yt} e^{-\alpha yt}}{e^{\alpha t} e^{-\alpha t}} = 1 = y^0.$$

The same property holds for $c(t)$. By Cline (1986, Theorem 4 (v)), as $t \to \infty$,

$$\mathbb{P} \left( |w_i| \ |\sum_{j=1}^{k} s_j \langle w_i^{(0)}, x_j \rangle I(\langle w_i^{(0)}, x_j \rangle > 0)| > e^t \right) = \overline{F * G}(t)$$

$$\sim C_\alpha^2 \mathbb{E}_{u \sim \Gamma_0} \left( |\sum_{j=1}^{k} s_j \langle u, x_j \rangle I(\langle u, x_j \rangle > 0)|^\alpha \right) \alpha t e^{-\alpha t},$$

as $t \to \infty$. Thus, for $r \to \infty$,

$$r^\alpha \mathbb{P}\left(|w_i| \ |\sum_{j=1}^k s_j \langle w_i^{(0)}, x_j \rangle I(\langle w_i^{(0)}, x_j \rangle > 0)| > r\right)$$

$$\sim C_\alpha^2 \mathbb{E}_{u \sim \Gamma_0}\left(|\sum_{j=1}^k s_j \langle u, x_j \rangle I(\langle u, x_j \rangle > 0)|^\alpha\right)\alpha \log r.$$

Let $\tilde{L}(r) = C_\alpha^2 \mathbb{E}_{u \sim \Gamma_0}\left(|\sum_{j=1}^k s_j \langle u, x_j \rangle I(\langle u, x_j \rangle > 0)|^\alpha\right)\alpha \log r$. Since the distribution of $w_i \sum_{j=1}^k s_j \langle w_i^{(0)}, x_j \rangle I(\langle w_i^{(0)}, x_j \rangle > 0)$ is symmetric, then we can write that

$$\frac{1}{a_m} \sum_{i=1}^m w_i \sum_{j=1}^k s_j \langle w_i^{(0)}, x_j \rangle I(\langle w_i^{(0)}, x_j \rangle > 0),$$

as a sequence of random variables in $m$, converges in distribution, as $m \to +\infty$, to a random variable with symmetric $\alpha$-stable law with scale 1 provided $(a_m)_{m \geq 1}$ satisfies

$$\frac{m\tilde{L}(a_m)}{a_m^\alpha} \to C_\alpha$$

as $m \to \infty$. The condition is satisfied if

$$a_m = \left(C_\alpha \mathbb{E}_{u \sim \Gamma_0}\left(|\sum_{j=1}^k s_j \langle u, x_j \rangle I(\langle u, x_j \rangle > 0)|^\alpha\right)m \log m\right)^{1/\alpha}.$$

It follows that

$$\frac{1}{(m \log m)^{1/\alpha}} \sum_{i=1}^m w_i \sum_{j=1}^k s_j \langle w_i^{(0)}, x_j \rangle I(\langle w_i^{(0)}, x_j \rangle > 0),$$

as a sequence of random variables in $m$, converges in distribution, as $m \to +\infty$, to a random variable with symmetric $\alpha$-stable distribution with scale of the form

$$\left(C_\alpha \mathbb{E}_{u \sim \Gamma_0}\left(|\sum_{j=1}^k s_j \langle u, x_j \rangle I(\langle u, x_j \rangle > 0)|^\alpha\right)\right)^{1/\alpha}.$$

Since this holds for every vector $s$, then

$$\frac{1}{(m \log m)^{1/\alpha}} \sum_{i=1}^m w_i [\langle w_i^{(0)}, x_j \rangle I(\langle w_i^{(0)}, x_j \rangle > 0)]_j,$$

as a sequence of random variables in $m$, converges in distribution, as $m \to +\infty$, to a random vector with symmetric $\alpha$-stable law with the spectral measure

$$\Gamma_X = \frac{1}{2}C_\alpha \mathbb{E}_{u \sim \Gamma_0}\Bigg(\|[\langle u, x_j \rangle I(\langle u, x_j \rangle > 0)]_j\|^\alpha$$

$$\delta\left(\frac{[\langle u, x_j \rangle I(\langle u, x_j \rangle > 0)]_j}{\|[\langle u, x_j \rangle I(\langle u, x_j \rangle > 0)]_j\|}\right) + \delta\left(-\frac{[\langle u, x_j \rangle I(\langle u, x_j \rangle > 0)]_j}{\|[\langle u, x_j \rangle I(\langle u, x_j \rangle > 0)]_j\|}\right)\Bigg).$$

Since $\Gamma_0 = \frac{1}{2}\sum_{i=1}^d (\delta(e_i) + \delta(-e_i))$, where $e_{ij} = 1$ if $j = i$ and 0 otherwise, then

$$\Gamma_X = \frac{C_\alpha}{4} \sum_{i=1}^d \Bigg(\|[x_{ji}I(x_{ji} > 0)]_j\|^\alpha \left(\delta\left(\frac{[x_{ji}I(x_{ji} > 0)]_j}{\|[x_{ji}I(x_{ji} > 0)]_j\|}\right) + \delta\left(-\frac{[x_{ji}I(x_{ji} > 0)]_j}{\|[x_{ji}I(x_{ji} > 0)]_j\|}\right)\right)$$

$$+\|[x_{ji}I(x_{ji} < 0)]_j\|^\alpha \left(\delta\left(\frac{[x_{ji}I(x_{ji} < 0)]_j}{\|[x_{ji}I(x_{ji} < 0)]_j\|}\right) + \delta\left(-\frac{[x_{ji}I(x_{ji} < 0)]_j}{\|[x_{ji}I(x_{ji} < 0)]_j\|}\right)\right)\Bigg).$$

## A.2 Proof of Theorem 3.1

To simplify the notation, we set in this section: $w := w(0)$, $w^{(0)} := w^{(0)}(0)$, $W := W(0)$, $\tilde{H}_m^{(1)} := \tilde{H}_m^{(1)}(W(0), X)$ and $\tilde{H}_m^{(2)} := \tilde{H}_m^{(2)}(W(0), X)$, with $\tilde{H}_m^{(1)}(W, X)$ and $\tilde{H}_m^{(2)}(W, X)$ defined in (12) and (13). The proof of Theorem 3.1 is split into several steps.

**Lemma A.1.** *If $m \to +\infty$ then*

$$\tilde{H}_m^{(1)} \xrightarrow{w} \tilde{H}_1^*(\alpha),$$

*where $\tilde{H}_1^*(\alpha)$ is an $(\alpha/2)$-Stable positive semi-definite random matrix with spectral measure*

$$\Gamma_1^* = C_{\alpha/2} \sum_{u \in \{0,1\}^k} \mathbb{P}(w_i^{(0)} \in B_u)(\sum_{j,j'} \langle x_j, x_{j'}\rangle^2 u_j u_{j'})^{\alpha/4} \delta\left(\frac{[\langle x_j, x_{j'}\rangle u_j u_{j'}]_{j,j'}}{(\sum_{j,j'} \langle x_j, x_{j'}\rangle^2 u_j u_{j'})^{1/2}}\right),$$

*where, for every $u \in \{0,1\}^k$, $B_u = \{v \in \mathbb{R}^d : \langle v, x_j\rangle > 0$ if $u_j = 1, \langle v, x_j\rangle \leq 0$ if $u_j = 0, j = 1, \ldots, k\}$, and $C_{\alpha/2}$ is the constant defined in (4).*

*Proof.* Since $\tilde{H}_m^{(1)}$ is symmetric, is is sufficient to show that, for every $k$-dimensional vector $s$,

$$s^T \tilde{H}_m^{(1)} s \xrightarrow{w} s^T \tilde{H}_1^*(\alpha)s.$$

We first prove that the functions defined, for $t \in (-\infty, +\infty)$, by $\overline{F}(t) = \mathbb{P}\left(w_i^2 > e^t\right)$, and

$$\overline{G}(t) = \mathbb{P}\left(\sum_{j,j'=1}^k s_j s_{j'} \langle x_j, x_{j'}\rangle I(\langle w_i^{(0)}, x_j\rangle > 0)I(\langle w_i^{(0)}, x_{j'}\rangle > 0) > e^t\right)$$

$$= \mathbb{P}\left(\|\sum_{j=1}^k s_j x_j I(\langle w_i^{(0)}, x_j\rangle > 0)\|^2 > e^t\right)$$

satisfy the assumptions of Cline (1986, Lemma 1). Indeed, $F$ has exponentail tails with rate $\alpha/2$, since by the properties of the stable law,

$$\lim_{t \to \infty} \frac{\overline{F}(t-u)}{\overline{F}(t)} = \lim_{t \to \infty} \frac{\mathbb{P}(|w_i| > e^{(t-u)/2})}{\mathbb{P}(|w_i| > e^{t/2})} = e^{\alpha u/2}.$$

Moreover, for any $\gamma$,

$$m_G(\gamma) = \int_0^\infty e^{\gamma u} G(du) = \mathbb{E}\left(\|\sum_{j=1}^k s_j x_j I(\langle w_i^{(0)}, x_j\rangle > 0)\|^\gamma\right) < \infty.$$

By Cline (1986), Section 5 and Lemma 1, as $t \to \infty$,

$$\mathbb{P}\left(w_i^2 \sum_{j,j'=1}^k s_j s_{j'} \langle x_j, x_{j'}\rangle I(\langle w_i^{(0)}, x_j\rangle > 0)I(\langle w_i^{(0)}, x_{j'}\rangle > 0) > e^t\right)$$

$$= \overline{F * G}(t) \sim m_G(\alpha/2)\overline{F}(t)$$

$$\sim C_{\alpha/2}(e^t)^{-\alpha/2}\mathbb{E}\left(\left(\sum_{j,j'=1}^k s_j s_{j'} \langle x_j, x_{j'}\rangle I(\langle w_i^{(0)}, x_j\rangle > 0)I(\langle w_i^{(0)}, x_{j'}\rangle > 0)\right)^{\alpha/2}\right).$$

By the properties of the stable law,

$$s^T \tilde{H}_m^{(1)} s = \frac{1}{m^{2/\alpha}} \sum_{i=1}^m w_i^2 \sum_{j,j'=1}^k s_j s_{j'} \langle x_j, x_{j'}\rangle I(\langle w_i^{(0)}, x_j\rangle > 0)I(\langle w_i^{(0)}, x_{j'}\rangle > 0)$$

converges weakly, as $m \to \infty$, to a totally skewed to the right, $\alpha/2$-stable random variable, with scale parameter $\mathbb{E}\big(|\sum_{j,j'=1}^k s_j s_{j'}\langle x_j, x_{j'}\rangle I(\langle w_i^{(0)}, x_j\rangle > 0) I(\langle w_i^{(0)}, x_{j'}\rangle > 0)|^{\alpha/2}\big)^{2/\alpha}$. Hence, for every $t \in \mathbb{R}$, as $m \to \infty$,

$$\mathbb{E}\left(\exp(its^T \tilde{H}_m^{(1)} s)\right)$$

$$\to \exp\left(-|t|^{\alpha/2}\mathbb{E}\big(|\sum_{j,j'=1}^k s_j s_{j'}\langle x_j, x_{j'}\rangle I(\langle w_i^{(0)}, x_j\rangle > 0) I(\langle w_i^{(0)}, x_{j'}\rangle > 0)|^{\alpha/2}\big)\big(1 - i \operatorname{sign} u \tan(\pi\alpha/4)\big)\right)$$

$$= \exp\left(-\int_{\mathbb{S}^{k^2-1}} |\sum_{j,j'} t s_j s_{j'} v_{j,j'}|^{\alpha/2}\big(1 - i \operatorname{sign}(t \sum_{j,j'} s_j s_{j'} v_{j,j'}) \tan(\pi\alpha/4)\Gamma_1^*(dv)\right),$$

where

$$\Gamma_1^* = C_{\alpha/2}\mathbb{E}\left(\|[\langle x_j, x_{j'}\rangle I(\langle w_i^{(0)}, x_j\rangle > 0) I(\langle w_i^{(0)}, x_{j'}\rangle > 0)]_{j,j'}\|_F^{\alpha/2}\right.$$

$$\left.\cdot \delta\left(\frac{[\langle x_j, x_{j'}\rangle I(\langle w_i^{(0)}, x_j\rangle > 0) I(\langle w_i^{(0)}, x_{j'}\rangle > 0)]_{j,j'}}{\|[\langle x_j, x_{j'}\rangle I(\langle w_i^{(0)}, x_j\rangle > 0) I(\langle w_i^{(0)}, x_{j'}\rangle > 0)]_{j,j'}\|_F}\right)\right).$$

It follows that, as $m \to +\infty$,

$$\tilde{H}_m^{(1)} \xrightarrow{\text{w}} \tilde{H}_1^*(\alpha),$$

where $\tilde{H}_1^*(\alpha)$ is an $(\alpha/2)$-Stable random matrix with spectral measure $\Gamma_1^*$ of the form

$$\Gamma_1^* = C_{\alpha/2} \sum_{u \in \{0,1\}^k} \mathbb{P}(w_i^{(0)} \in B_u)(\sum_{j,j'} \langle x_j, x_{j'}\rangle^2 u_j u_{j'})^{\alpha/4}\delta\left(\frac{[\langle x_j, x_{j'}\rangle u_j u_{j'}]_{j,j'}}{(\sum_{j,j'} \langle x_j, x_{j'}\rangle^2 u_j u_{j'})^{1/2}}\right).$$

We will now prove that $\tilde{H}_1^*(\alpha)$ is positive semi-definite. By definition, $\tilde{H}_m^{(1)}(\omega)$ is positive semi-definite for every $\omega$ and every $m$. By Portmanteau Theorem, for every vector $u \in \mathbb{S}^{k-1}$,

$$\mathbb{P}\left(u^T \tilde{H}_1^*(\alpha)u \geq 0\right) \geq \limsup_m \mathbb{P}\left(u^T \tilde{H}_m^{(1)} u \geq 0\right) = 1.$$

Let $\mathcal{A}$ be a countable dense subset of $\mathbb{S}^{k-1}$. Then, with probability one, $a^T \tilde{H}_1^*(\alpha)a \geq 0$ for every $a \in \mathcal{A}$. By continuity, this implies that the same property holds true with probability one for every $u \in \mathbb{S}^{k-1}$, which proves that $\tilde{H}_1^*(\alpha)$ is almost surely positive semi-definite. By eventually modifying $\tilde{H}_1^*(\alpha)$ on a null set, we obtain a positive semi-definite random matrix. $\qquad\square$

**Lemma A.2.** *If $m \to +\infty$ then*

$$\tilde{H}_m^{(2)} \xrightarrow{w} \tilde{H}_2^*(\alpha),$$

*where $\tilde{H}_2^*(\alpha)$ is an $(\alpha/2)$-Stable positive semi-definite random matrix with spectral measure*

$$\Gamma_2^* = C_{\alpha/2} \sum_{u \in \{0,1\}^k} \sum_{\{i:\{e_i, -e_i\}\cap B_u \neq \emptyset\}} (\sum_j x_{ji}^2 u_j)^{\alpha/2}\delta\left(\frac{[x_{ji} u_j x_{j'i} u_{j'}]_{j,j'}}{\sum_j x_{ji}^2 u_j}\right),$$

*where $B_u = \{v \in \mathbb{R}^d : \langle v, x_j\rangle > 0 \text{ if } u_j = 1, \langle v, x_j\rangle \leq 0 \text{ if } u_j = 0, j = 1, \ldots, k\}$, $e_i$ is a $d$-dimensional vector satisfying $e_{ij} = 1$ if $j = i$, and $e_{ij} = 0$ if $j \neq i$ $(i, j = 1, \ldots, d)$, and $C_{\alpha/2}$ is the constant defined in (4).*

*Proof.* By the properties of the multivariate stable distribution (see Appendix B), it is sufficient to show that

$$\mathbb{P}\left(\frac{\left[\langle w_1^{(0)}, x_j\rangle\langle w_1^{(0)}, x_{j'}\rangle I(\langle w_1^{(0)}, x_j\rangle > 0)I(\langle w_1^{(0)}, x_{j'}\rangle > 0)\right]_{j,j'}}{\|\left[\langle w_1^{(0)}, x_j\rangle\langle w_1^{(0)}, x_{j'}\rangle I(\langle w_1^{(0)}, x_j\rangle > 0)I(\langle w_1^{(0)}, x_{j'}\rangle > 0)\right]_{j,j'}\|_F} \in \cdot,\right.$$
$$\left.\|\left[\langle w_1^{(0)}, x_j\rangle\langle w_1^{(0)}, x_{j'}\rangle I(\langle w_1^{(0)}, x_j\rangle > 0)I(\langle w_1^{(0)}, x_{j'}\rangle > 0)\right]_{j,j'}\|_F > r\right)$$
$$\sim C_{\alpha/2} r^{-\alpha/2} \Gamma_2^*(\cdot),$$

as $r \to +\infty$. We can write that

$$\mathbb{P}\left(\frac{\left[\langle w_1^{(0)}, x_j\rangle\langle w_1^{(0)}, x_{j'}\rangle I(\langle w_1^{(0)}, x_j\rangle > 0)I(\langle w_1^{(0)}, x_{j'}\rangle > 0)\right]_{j,j'}}{\|\left[\langle w_1^{(0)}, x_j\rangle\langle w_1^{(0)}, x_{j'}\rangle I(\langle w_1^{(0)}, x_j\rangle > 0)I(\langle w_1^{(0)}, x_{j'}\rangle > 0)\right]_{j,j'}\|_F} \in \cdot,\right.$$
$$\left.\|\left[\langle w_1^{(0)}, x_j\rangle\langle w_1^{(0)}, x_{j'}\rangle I(\langle w_1^{(0)}, x_j\rangle > 0)I(\langle w_1^{(0)}, x_{j'}\rangle > 0)\right]_{j,j'}\|_F > r\right)$$
$$= \sum_{u \in \{0,1\}^k} \mathbb{P}\left(\frac{\left[\langle w_1^{(0)}, u_j x_j\rangle\langle w_1^{(0)}, u_{j'} x_{j'}\rangle\right]_{j,j'}}{\|\left[\langle w_1^{(0)}, u_j x_j\rangle\langle w_1^{(0)}, u_{j'} x_{j'}\rangle\right]_{j,j'}\|_F} \in \cdot,\right.$$
$$\left.\|\left[\langle w_1^{(0)}, u_j x_j\rangle\langle w_1^{(0)}, u_{j'} x_{j'}\rangle\right]_{j,j'}\|_F > r, w_1^{(0)} \in B_u\right).$$

For every $u \in \{0,1\}^k$, let $X_u$ be the $d \times k$ matrix, defined as

$$X_u = [x_{ji} u_j]_{j=1,\ldots,k, i=1,\ldots,d}.$$

Then we can write that

$$\mathbb{P}\left(\frac{\left[\langle w_1^{(0)}, u_j x_j\rangle\langle w_1^{(0)}, u_{j'} x_{j'}\rangle\right]_{j,j'}}{\|\left[\langle w_1^{(0)}, u_j x_j\rangle\langle w_1^{(0)}, u_{j'} x_{j'}\rangle\right]_{j,j'}\|_F} \in \cdot,\right.$$
$$\left.\|\left[\langle w_1^{(0)}, u_j x_j\rangle\langle w_1^{(0)}, u_{j'} x_{j'}\rangle\right]_{j,j'}\|_F > r, w_1^{(0)} \in B_u\right)$$
$$= \mathbb{P}\left(\frac{X_u^T w_1^{(0)} (w_1^{(0)})^T X_u}{(\operatorname{tr}(X_u^T (w_1^{(0)})^T w_1^{(0)} X_u X_u^T (w_1^{(0)})^T w_1^{(0)} X_u))^{1/2}} \in \cdot,\right.$$
$$\left.\operatorname{tr}(X_u^T (w_1^{(0)})^T w_1^{(0)} X_u X_u^T (w_1^{(0)})^T w_1^{(0)} X_u) > r^2, w_1^{(0)} \in B_u\right)$$
$$= \mathbb{P}\left(\frac{X_u^T (w_1^{(0)})^T w_1^{(0)} X_u}{w_1^{(0)} X_u X_u^T (w_1^{(0)})^T} \in \cdot, w_1^{(0)} X_u X_u^T (w_1^{(0)})^T > r, w_1^{(0)} \in B_u\right).$$

Notice that the maximum eigenvalue of the matrix $X_u X_u^T$ is smaller than or equal to $k$, since the norm of each column of $X_u$ is smaller than or equal to one. Then $w_1^{(0)} X_u X_u^T (w_1^{(0)})^T > r$ implies that $\|w_1^{(0)}\| > (r/k)^{1/2}$. We can therefore write that

$$\mathbb{P}\left(\frac{X_u^T (w_1^{(0)})^T w_1^{(0)} X_u}{w_1^{(0)} X_u X_u^T (w_1^{(0)})^T} \in \cdot, w_1^{(0)} X_u X_u^T (w_1^{(0)})^T > r, w_1^{(0)} \in B_u\right)$$
$$= \mathbb{P}\left(\frac{X_u^T (w_1^{(0)})^T w_1^{(0)} X_u}{w_1^{(0)} X_u X_u^T (w_1^{(0)})^T} \in \cdot, w_1^{(0)} X_u X_u^T (w_1^{(0)})^T > r, \|w_1^{(0)}\| > (r/k)^{1/2}, w_1^{(0)} \in B_u\right).$$

Since $B_u$ is a cone and the spectral measure of $w_1^{(0)}$ is given by $\sum_i (\delta(e_i) + \delta(-e_i))$, by the properties of the multivariate stable distribution, we can write that

$$\mathbb{P}\left(\frac{X_u^T (w_1^{(0)})^T w_1^{(0)} X_u}{w_1^{(0)} X_u X_u^T (w_1^{(0)})^T} \in \cdot, w_1^{(0)} X_u X_u^T (w_1^{(0)})^T > r, \|w_1^{(0)}\| > (r/k)^{1/2}, w_1^{(0)} \in B_u\right)$$

$$\sim C_{\alpha/2} r^{-\alpha/2} \sum_{\{i:\{e_1, -e_i\} \cap B_u \neq \emptyset\}} (\sum_{j=1}^k x_{ji}^2 u_j)^{\alpha/2} \delta\left(\frac{[x_{ji} x_{j'i} u_j u_{j'}]_{j,j'}}{\sum_j x_{ji}^2 u_j}\right),$$

as $r \to +\infty$. The proof that $\tilde{H}_2^*(\alpha)$ is positive semi-definite can be done by following the same line of reasoning as in the proof of Lemma A.1. $\qquad\square$

**Lemma A.3.** *As $m \to +\infty$, the probability distribution of $(\tilde{H}_m^{(1)}, \tilde{H}_m^{(1)})$ converges weakly to the law of independent stable random matrices, with spectral measures $\Gamma_1^*$ and $\Gamma_2^*$ as in (14) and (15), respectively.*

*Proof.* Since $\tilde{H}_m^{(1)}$ and $H_m^{(2)}$ converge marginally to $\alpha/2$-stable random matrices, by the properties of the multivariate stable distributions it is sufficient to show that they converge to stochastically independent random matrices. By Theorem B.1, we know that

$$n\mathbb{P}\Bigg(\|[w_i^2\langle x_j, x_{j'}\rangle I(\langle w_i^{(0)}, x_j\rangle > 0) I(\langle w_i^{(0)}, x_{j'}\rangle > 0)]_{j,j'}\|_F > n^{2/\alpha},$$
$$\|[\langle x_j, w_i^{(0)}\rangle\langle x_{j'}, w_i^{(0)}\rangle I(\langle w_i^{(0)}, x_j\rangle > 0) I(\langle w_i^{(0)}, x_{j'}\rangle > 0)]_{j,j'}\|_F > n^{2/\alpha}\Bigg)$$

and

$$n\mathbb{P}\left(\|[w_i^2\langle x_j, x_{j'}\rangle I(\langle w_i^{(0)}, x_j\rangle > 0) I(\langle w_i^{(0)}, x_{j'}\rangle > 0)]_{j,j'}\|_F > n^{2/\alpha}\right)$$

converge to finite limits, as $n \to \infty$. Hence, again by Theorem B.1, it is sufficient to show that

$$\lim_{n\to\infty} n\mathbb{P}\Bigg(\|[w_i^2\langle x_j, x_{j'}\rangle I(\langle w_i^{(0)}, x_j\rangle > 0) I(\langle w_i^{(0)}, x_{j'}\rangle > 0)]_{j,j'}\|_F > n^{2/\alpha},$$
$$\|[\langle x_j, w_i^{(0)}\rangle\langle x_{j'}, w_i^{(0)}\rangle I(\langle w_i^{(0)}, x_j\rangle > 0) I(\langle w_i^{(0)}, x_{j'}\rangle > 0)]_{j,j'}\|_F > n^{2/\alpha}\Bigg) = 0,$$

which ensures that the Lévy measure of the limit infinitely divisible distribution of $(\tilde{H}_m^{(1)}, \tilde{H}_m^{(2)})$ is the sum of a measure $\nu_1$ concentrated on the space spanned by the first $k^2$ coordinates and a measure $\nu_2$ on the space

spanned by the last $k^2$ coordinates. We can write that

$$n\mathbb{P}\Bigg(\|[w_i^2\langle x_j, x_{j'}\rangle I(\langle w_i^{(0)}, x_j\rangle > 0)I(\langle w_i^{(0)}, x_{j'}\rangle > 0)]_{j,j'}\|_F > n^{2/\alpha},$$

$$\|[\langle x_j, w_i^{(0)}\rangle\langle x_{j'}, w_i^{(0)}\rangle I(\langle w_i^{(0)}, x_j\rangle > 0)I(\langle w_i^{(0)}, x_{j'}\rangle > 0)]_{j,j'}\|_F > n^{2/\alpha}\Bigg)$$

$$= n\sum_{u\in\{0,1\}^k}\mathbb{P}(w_i^{(0)}\in B_u)$$

$$\mathbb{P}\Bigg(\|[w_i^2\langle x_j, x_{j'}\rangle u_j u_{j'}]_{j,j'}\|_F > n^{2/\alpha}, \|[\langle x_j, w_i^{(0)}\rangle\langle x_{j'}, w_i^{(0)}\rangle u_j u_{j'}]_{j,j'}\|_F > n^{2/\alpha} \mid w_i^{(0)}\in B_u\Bigg)$$

$$= n\sum_{u\in\{0,1\}^k}\mathbb{P}(w_i^{(0)}\in B_u)\mathbb{P}\Bigg(\|[\langle x_j, w_i^{(0)}\rangle\langle x_{j'}, w_i^{(0)}\rangle u_j u_{j'}]_{j,j'}\|_F > n^{2/\alpha} \mid w_i^{(0)}\in B_u\Bigg)$$

$$\mathbb{P}\Bigg(\|[w_i^2\langle x_j, x_{j'}\rangle u_j u_{j'}]_{j,j'}\|_F > n^{2/\alpha}\Bigg)$$

$$= \sum_{u\in\{0,1\}^k} n\mathbb{P}\Bigg(\|[\langle x_j, w_i^{(0)}\rangle\langle x_{j'}, w_i^{(0)}\rangle u_j u_{j'}]_{j,j'}\|_F > n^{2/\alpha}, w_i^{(0)}\in B_u\Bigg)$$

$$\mathbb{P}\Bigg(\|[w_i^2\langle x_j, x_{j'}\rangle u_j u_{j'}]_{j,j'}\|_F > n^{2/\alpha}\Bigg) \to 0,$$

as $n\to\infty$. $\qquad\square$

Now, we are in the position of proving Theorem 3.1. By Lemma A.1, Lemma A.1, Lemma A.3, and the properties of stable distributions, $\tilde{H}_m(W(0), X)$ converges in distribution to a positive semi-definite random matrix, with $(\alpha/2)$-stable distribution, and spectral measure $\Gamma_1^* + \Gamma_2^*$. This completes the proof of Theorem 3.1.

### A.3 Proof of Proposition 3.1

To simplify the notation, we set in this section: $w := w(0)$, $w^{(0)} := w^{(0)}(0)$, $W := W(0)$, $\tilde{H}_m^{(1)} := \tilde{H}_m^{(1)}(W(0), X)$ and $\tilde{H}_m^{(2)} := \tilde{H}_m^{(2)}(W(0), X)$, with $\tilde{H}_m^{(1)}(W, X)$ and $\tilde{H}_m^{(2)}(W, X)$ defined in (12) and (13).

From (11), $\tilde{H}_m(W(0), X))$ is the sum of two positive semi-definite random matrices, $\tilde{H}_m^{(1)}$ and $\tilde{H}_m^{(2)}$. The following results show that for every $\delta > 0$, there exist $\lambda_1 > 0$ and $\lambda_2 > 0$ such that, for $m$ sufficiently large, with probability at least $1 - \delta$

$$\lambda_{\min}(\tilde{H}_m^{(i)}) > \lambda_i.$$

with the large-width behaviour of $\tilde{H}_m^{(i)}$ being characterized in Lemma A.1 and Lemma A.2, through an $(\alpha/2)$-Stable limiting random matrix $\tilde{H}_i^*(\alpha)$ with spectral measure $\Gamma_i^*$ of the form (14) and (15). To prove that the minumum eigenvales of $\tilde{H}_m^{(1)}$ and $\tilde{H}_m^{(2)}$ are bounded away from zero, we first need to inspect the characteristics of the distributions of $\tilde{H}_1^*(\alpha)$ and of $\tilde{H}_2^*(\alpha)$. This is the content of Lemma A.4 and of Lemma A.6. Then, the results concerning the minumum eigenvalues of $\tilde{H}_m^{(1)}$ and $\tilde{H}_m^{(2)}$ are given in Lemma A.5 and Lemma A.7.

**Lemma A.4.** *Under the assumptions of Theorem 3.2, the distribution of the random matrix $\tilde{H}_1^*(\alpha)$ is absolutely continuous in the subspace of the symmetric positive semi-definite matrices with zero entries in the positions $(j, j')$ such that $\langle x_j, x_{j'}\rangle = 0$, with $j, j' \in \{1,\ldots,k\}$, with the topology of Frobenius norm.*

*Proof.* From Nolan (2010), it is sufficient to show that

$$\inf_{s\in\mathbb{S}_0^{k^2-1}}\int|\langle s, u\rangle|^{\alpha/2}\Gamma_1^*(\mathrm{d}u) \neq 0,$$

where $\Gamma_1^*$ is the spectral measure (14), $\mathbb{S}_0^{k^2-1}$ is the unit sphere in the space of the $k \times k$ symmetric matrices such that $s_{j,j'} = 0$ if $\langle x_j, x_{j'} \rangle = 0$, with the Frobenius metric. Now, since

$$\int |\langle s, u \rangle|^{\alpha/2} \Gamma_1^*(\mathrm{d}u)$$

$$= C_{\alpha/2} \mathbb{E} \left( | \sum_{j,j'} s_{j,j'} \langle x_j, x_{j'} \rangle I(\langle w_i^{(0)}, x_j \rangle > 0) I(\langle w_i^{(0)}, x_{j'} \rangle > 0)|^{\alpha/2} \right)$$

is a continuous function of $s$ that takes value in a compact set, then the minimum is attained. Thus it is sufficient to show that for every $s \in \mathbb{S}_0^{k^2-1}$,

$$\mathbb{E} \left( | \sum_{j,j'} s_{j,j'} \langle x_j, x_{j'} \rangle I\langle w_i^{(0)}, x_j \rangle > 0) I(\langle w_i^{(0)}, x_{j'} \rangle > 0)|^{\alpha/2} \right) \neq 0.$$

For every $j$ and every $u_j \in \{0, 1\}$, let $A_j^{u_j}$ be the event $(\langle w_i^{(0)}, x_j \rangle > 0)$ if $u_j = 1$ and its complement if $u_j = 0$. Then

$$\mathbb{E} \left( | \sum_{j,j'} s_{j,j'} \langle x_j, x_{j'} \rangle I(\langle w_i^{(0)}, x_j \rangle > 0) I(\langle w_i^{(0)}, x_{j'} \rangle > 0)|^{\alpha/2} \right)$$

$$= \sum_{u_1,\ldots,u_k} \mathbb{P}(A_1^{u_1} \cap \cdots \cap A_k^{u_k}) | \sum_{j,j'} u_j u_{j'} s_{j,j'} \langle x_j, x_{j'} \rangle|^{\alpha/2}.$$

Since $x_1, \ldots, x_k$ are linearly independent, then for every $u_1, \ldots, u_k$, $\mathbb{P}(A_1^{u_1} \cap \ldots, A_k^{u_k}) > 0$. To prove it, assume, without loss of generality, that $u_i = 1$ for every $i$. Since $x_1, \ldots, x_k$ are linearly independent, then we can complete the matrix $X = [x_1 \ \ldots \ x_k]$ by adding $k - d$ columns in such a way that the completed matrix $\tilde{X}$ is non-singular. For every $d$-dimensional vector $v$ such that $v_1 > 0, \ldots, v_k > 0$ there exists a vector $u$ such that $u = (\tilde{X}^T)^{-1} v$. Thus,

$$\{u \in \mathbb{R}^d : \langle u, x_1 \rangle > 0, \ldots, \langle u, x_k \rangle > 0\} = \{(\tilde{X}^T)^{-1} v : v_1 > 0, \ldots, v_k > 0\}$$

is an open non-empty set. Since $w_i^{(0)}$ has independent and identically distributed components, with stable distribution, then

$$\mathbb{P} \left( w_i^{(0)} \in \{(\tilde{X})^{-1} v : v_1 > 0, \ldots, v_k > 0\} \right) > 0.$$

This concludes the proof that $\mathbb{P}(A_1^{u_1} \cap \ldots, A_k^{u_k}) > 0$ for every $(u_1, \ldots, u_k) \in \{0, 1\}^k\}$. It follows that $\int |\langle s, u \rangle|^{\alpha/2} \Gamma_1^*(du)$ is zero if and only if, for every $(u_1, \ldots, u_k) \in \{0, 1\}^k$, it holds

$$\sum_{j,j'} u_j, u_{j'} \langle x_j, x_{j'} \rangle s_{j,j'} = 0.$$

The only solution of the above system of equations in the space of symmetric matrices $s$ such that $s_{j,j'} = 0$ if $\langle x_j, x_{j'} \rangle = 0$ is $s = 0$, which is not consistent with $\|s\|_F = 1$. $\square$

We observe that the space of the symmetric positive semi-definite matrices with zeros in the entries $(j, j')$ such that $\langle x_j, x_{j'} \rangle = 0$ contains all the matrices with non-zero diagonal element since $\langle x_j, x_j \rangle = 1 \neq 0$ for every index $j$.

**Lemma A.5.** *Under the assumptions of Theorem 3.2, for every $\delta > 0$ there exists $\lambda_1 > 0$ such that with probability at least $1 - \delta$*

$$\lambda_{min}(\tilde{H}_1^*(\alpha)) > \lambda_1.$$

*Proof.* Since the distribution of $\tilde{H}_1^*(\alpha)$ is absolutely continuous in the space of symmetric positive semi-definite matrices with zero entries in the positions $j, j'$ such that $\langle x, x_{j'} \rangle = 0$, and since this space contains all the symmetric positive semi-definite matrices with non-zero diagonal entries, then we can write that $\mathbb{P}(\det(\tilde{H}_1^*(\alpha)) = 0) = 0$. Moreover, since $\tilde{H}_1^*(\alpha)$ is positive semi-definite, then $\mathbb{P}(\lambda_{\min}(\tilde{H}_1^*(\alpha)) > 0) = 1$. Thus, for every $\delta > 0$, the exists $\lambda_1 > 0$ such that $\mathbb{P}(\lambda_{\min}(\tilde{H}_1^*(\alpha)) > \lambda_1) > 1 - \delta$. $\square$

**Lemma A.6.** *Under the assumptions of Theorem 3.2, the distribution of the random matrix $\tilde{H}_2^*(\alpha)$ is absolutely continuous in the subspace of the symmetric positive semi-definite matrices, with the topology of Frobenius norm.*

*Proof.* From Nolan (2010), it is sufficient to show that

$$\inf_{s \in \mathbb{S}^{k^2-1}} \int |\langle s, u \rangle|^{\alpha/2} \Gamma_2^*(\mathrm{d}u) \neq 0,$$

where $\Gamma_2^*$ is the spectral measure (15), $\mathbb{S}^{k^2-1}$ is the unit sphere in the space of the $k \times k$ symmetric positive semi-definite matrices, with the Frobenius norm. For every $u \in \{0,1\}^k$, let $B_u = \{v \in \mathbb{R}^d : \langle v, x_j \rangle > 0 \text{ if } u_j = 1, \langle v, x_j \rangle \leq 0 \text{ if } u_j = 0\}$. Moreover, for every $i = 1, \ldots, k$, let $e_i$ be a $d$-dimensional random vector satisfying $e_{ij} = 1$ for $j = i$ and $e_{ij} = 0$ for $j \neq i$. Finally, let $C_{\alpha/2}$ be the constant defined in (4). Then

$$\int |\langle s, u \rangle|^{\alpha/2} \Gamma_2^*(\mathrm{d}u) = C_{\alpha/2} |\sum_{j,j'} s_{j,j'} \sum_{u \in \{0,1\}^k} \sum_{\{i:\{e_i,-e_i\} \cap B_u \neq \emptyset\}} x_{ji} u_j x_{j'i} u_{j'}|^{\alpha/2}.$$

Since $\sum_{j,j'} s_{j,j'} \sum_{u \in \mathcal{U}} \sum_E z_{u,i} x_{ji} u_j x_{j'i} u_{j'}$ is continuous as a function of $s$ and $s$ takes values in a compact set, then the minimum is attained. Thus it is sufficient to show that for every $s \in \mathbb{S}^{k^2-1}$,

$$\sum_{u \in \{0,1\}^k} \sum_{\{i:\{e_i,-e_i\} \cap B_u \neq \emptyset\}} \sum_{j,j'} s_{j,j'} x_{ji} u_j x_{j'i} u_{j'} \neq 0.$$

Since $\|s\|_F = 1$, then $s$ is not the null matrix. Hence there exist $c > 0$, a vector $a$ with $\|a\| = 1$ and a positive semi-definite, symmetric matrix $s'$ such that

$$s = c a a^T + s'.$$

Since $B_u \cap B_{u'} = \emptyset$, when $u \neq u'$, then, for every $i = 1, \ldots, d$ and $j = 1, \ldots, k$, there exists one and only one $u \in \{0,1\}^k$ such that $u_j = 1$ and $\{e_i, -e_i\} \cap B_u \neq \emptyset$. Then we can write that

$$\sum_{u \in \{0,1\}^k} \sum_{\{i:\{e_i,-e_i\} \cap B_u \neq \emptyset\}} \sum_{j,j'} s_{j,j'} x_{ji} u_j x_{j'i} u_{j'}$$

$$\geq c \sum_{u \in \{0,1\}^k} \sum_{\{i:\{e_i,-e_i\} \cap B_u \neq \emptyset\}} (\sum_j a_j x_{ji} u_j)^2$$

$$= \sum_{i=1}^d \left( (\sum_{j=1}^k a_j x_{ji})^2 \sum_{\{u:\{e_i,-e_i\} \cap B_u \neq \emptyset\}} u_j \right)$$

$$= \sum_{i=1}^d (\sum_{j=1}^k a_j x_{ji})^2,$$

which is strictly positive, since the $x_j$ are linearly independent, and $\|a\| = 1$. This concludes the proof. $\square$

**Lemma A.7.** *Under the assumptions of Theorem 3.2, for every $\delta > 0$ there exists $\lambda_2 > 0$ such that with probability at least $1 - \delta$*

$$\lambda_{min}(\tilde{H}_2^*(\alpha)) > \lambda_2.$$

*Proof.* Since the distribution of $\tilde{H}_2^*(\alpha)$ is absolutely continuous in the space of symmetric positive semi-definite matrices then we can write that $\mathbb{P}(\det(\tilde{H}_2^*(\alpha)) = 0) = 0$. Moreover, since $\tilde{H}_2^*(\alpha)$ is positive semi-definite, then $\mathbb{P}(\lambda_{\min}(\tilde{H}_2^*(\alpha)) > 0) = 1$. Thus, for every $\delta > 0$, the exists $\lambda_2 > 0$ such that $\mathbb{P}(\lambda_{\min}(\tilde{H}_2^*(\alpha)) > \lambda_2) > 1 - \delta$. □

Now, we are in the position of proving Proposition 3.1. Let $\delta > 0$ be a fixed number. By Lemmas A.5 and A.7, there exist $\lambda_1 > 0$ and $\lambda_2 > 0$ such that, for $i = 1, 2$, $\mathbb{P}(\lambda_{\min}(\tilde{H}_i^*(\alpha)) > \lambda_i) \geq 1 - \delta/2$. Since the minimum eigenvalue map is continuous with respect to Frobenius norm then, by Portmanteau theorem, for $i = 1, 2$,

$$\liminf_m \mathbb{P}(\lambda_{\min}(\tilde{H}_m^{(i)}(W(0), X)) > \lambda_i) \geq \mathbb{P}(\lambda_{\min}(\tilde{H}_i^*(\alpha)) > \lambda_i) \geq 1 - \delta/2.$$

Let $\lambda_0 = \lambda_1 + \lambda_2$. Since the minimum eigenvalue of a sum of symmetric, positive semi-definite matrices is greater than or equal to the sum of the eigenvalues of the two matrices (see Horn and Johnson (1985) Theorem 4.3.1), then we can write that

$$\liminf_m \mathbb{P}(\lambda_{\min}(\tilde{H}_m(W(0), X)) > \lambda_0)$$
$$\geq \liminf_m \mathbb{P}(\lambda_{\min}(\tilde{H}_m^{(1)}(W(0), X)) + \lambda_{\min}(\tilde{H}_m^{(2)}(W(0), X)) > \lambda_0)$$
$$\geq \liminf_m \mathbb{P}(\cap_{i=1,2}(\lambda_{\min}(\tilde{H}_m^{(i)}(W(0), X)) > \lambda_i))$$
$$\geq 1 - \limsup_m \left( \sum_{i=1}^{2} \mathbb{P}(\lambda_{\min}(\tilde{H}_m^{(i)}(W(0), X)) \leq \lambda_i) \right)$$
$$\geq 1 - \delta,$$

thus completing the proof of Proposition 3.1.

### A.4 Proof of Proposition 3.2

Before proving Proposition 3.2, we give some preliminary results.

**Lemma A.8.** *Let $\gamma \in (0, 1)$ and $c > 0$ be fixed numbers. For every $\delta > 0$ the following property holds true, for $m$ sufficiently large, with probability at least $1 - \delta$:*

$$(\log m)^{2/\alpha} \left\| \frac{\partial \tilde{f}_m}{\partial w}(W, x_j; \alpha) - \frac{\partial \tilde{f}_m}{\partial w}(W(0), x_j; \alpha) \right\|_F^2 < cm^{-2\gamma/\alpha},$$

*for every $W$ such that $\|W - W(0)\|_F \leq (\log m)^{2/\alpha}$ and every NN's input $x_j$, with $j = 1, \ldots, k$.*

*Proof.* For a fixed $W(0)$, let $W$ be such that $\|W - W(0)\|_F \leq (\log m)^{2/\alpha}$. Then it holds $\|w^{(0)} - w^{(0)}(0)\|_F^2 \leq \|W - W(0)\|_F^2 \leq (\log m)^{4/\alpha}$. Accordingly, we can write the following

$$(\log m)^{2/\alpha} \left\| \frac{\partial \tilde{f}_m}{\partial w}(W, x_j; \alpha) - \frac{\partial \tilde{f}_m}{\partial w}(W(0), x_j; \alpha) \right\|_F^2$$
$$\leq \frac{1}{m^{2/\alpha}} \sum_{i=1}^{m} \left( \langle w_i^{(0)}, x_j \rangle I(\langle w_i^{(0)}, x_j \rangle > 0) - \langle w_i^{(0)}(0), x_j \rangle I(\langle w_i^{(0)}(0), x_j \rangle > 0) \right)^2$$
$$\leq \frac{2}{m^{2/\alpha}} \sum_{i=1}^{m} \left( \langle w_i^{(0)}, x_j \rangle - \langle w_i^{(0)}(0), x_j \rangle \right)^2 I(\langle w_i^{(0)}, x_j \rangle > 0)$$
$$+ \frac{2}{m^{2/\alpha}} \sum_{i=1}^{m} \langle w_i^{(0)}(0), x_j \rangle^2 \left( I(\langle w_i^{(0)}, x_j \rangle > 0) - I(\langle w_i^{(0)}(0), x_j \rangle > 0) \right)^2.$$

We will bound the two terms of the sum separately. First, we define $r_i = |\langle w_i^{(0)} - w_i^{(0)}(0), x_j \rangle|$ for $i = 1, \ldots, m$. Then, we can write that

$$\sum_{i=1}^{m} r_i^2 \leq \sum_{i=1}^{m} \|w_i^{(0)} - w_i^{(0)}(0)\|^2 \cdot \|x_j\|^2 \leq \|w^{(0)} - w^{(0)}(0)\|_F^2 \leq (\log m)^{4/\alpha}.$$

Since $\gamma < 1$,

$$\frac{2}{m^{2/\alpha}} \sum_{i=1}^{m} \left( \langle w_i^{(0)}, x_j \rangle - \langle w_i^{(0)}(0), x_j \rangle \right)^2 I(\langle w_i^{(0)}, x_j \rangle > 0)$$

$$\leq 2m^{-2/\alpha} (\log m)^{4/\alpha} < \frac{c}{4} m^{-2\gamma/\alpha},$$

for $m$ sufficiently large. In order to bound the second term, we observe that the following set

$$\{w^{(0)}(0) : \exists w^{(0)} s.t. |\langle w_i^{(0)} - w_i^{(0)}(0), x_j \rangle| = r_i, \ I(\langle w^{(0)}, x_j \rangle > 0) \neq I(\langle w^{(0)}(0), x_j \rangle > 0)\}$$

is included in the set $\{w_i^{(0)}(0) : |\langle w_i^{(0)}(0), x_j \rangle| \leq r_i\}$. Therefore, we can write that

$$\sup_{\sum_i r_i^2 \leq \log m} \sup_{|w_i^{(0)} - w_i^{(0)}(0)| \leq r_i} \frac{2}{m^{2/\alpha}} \sum_{i=1}^{m} \langle w_i^{(0)}(0), x_j \rangle^2 \left( I(\langle w_i^{(0)}, x_j \rangle > 0) - I(\langle w_i^{(0)}(0), x_j \rangle > 0) \right)^2$$

$$\leq \sup_{\sum_i r_i^2 \leq \log m} \sup_{|w_i^{(0)} - w_i^{(0)}(0)| \leq r_i} \frac{2}{m^{2/\alpha}} \sum_{i=1}^{m} \langle w_i^{(0)}(0), x_j \rangle^2 I(\langle w_i^{(0)}(0), x_j \rangle < r_i)$$

$$\leq \sup_{\sum_i r_i^2 \leq \log m} \sup_{|w_i^{(0)} - w_i^{(0)}(0)| \leq r_i} \frac{2}{m^{2/\alpha}} \sum_{i=1}^{m} r_i^2$$

$$\leq \frac{1}{m^{2/\alpha}} (\log m)^{4/\alpha} < \frac{c}{4} m^{-2\gamma/\alpha},$$

for $m$ sufficiently large. $\qquad\square$

**Lemma A.9.** *For every $\delta > 0$ there exist $\lambda > 0$ such that the following two properties hold true, for $m$ sufficiently large, with a probability at least $1 - \delta$:*

*i)*

$$\|\tilde{H}_m^{(2)}(W, X) - \tilde{H}_m^{(2)}(W(0), X)\|_F < \lambda m^{-\gamma/\alpha};$$

*ii)*

$$\lambda_{min}(\tilde{H}_m(W, X)) > \frac{\lambda}{2};$$

*for every $W$ such that $\|W - W(0)\|_F \leq (\log m)^{2/\alpha}$.*

*Proof.* By Lemma A.7, for every $\delta > 0$ there exists $\lambda$ such that

$$\lambda_{\min}(\tilde{H}_2^*(\alpha)) > \lambda$$

with probability at least $1 - \delta/2$. For every vector $W$, we can write that

$$
\begin{aligned}
&|\tilde{H}_m^{(2)}(W, X)[i,j] - \tilde{H}_m^{(2)}(W(0), X)[i,j]| \\
&= (\log m)^{2/\alpha} \left| \left\langle \frac{\partial \tilde{f}_m}{\partial w}(W, x_i; \alpha), \frac{\partial \tilde{f}_m}{\partial w}(W, x_j; \alpha) \right\rangle - \left\langle \frac{\partial \tilde{f}_m}{\partial w}(W(0), x_i; \alpha), \frac{\partial \tilde{f}_m}{\partial w}(W(0), x_j; \alpha) \right\rangle \right| \\
&\leq (\log m)^{2/\alpha} \left\| \frac{\partial \tilde{f}_m}{\partial w}(W, x_i; \alpha) \right\|_F \left\| \frac{\partial \tilde{f}_m}{\partial w}(W, x_j; \alpha) - \frac{\partial \tilde{f}_m}{\partial w}(W(0), x_j; \alpha) \right\|_F \\
&\quad + (\log m)^{2/\alpha} \left\| \frac{\partial \tilde{f}_m}{\partial w}(W(0), x_j; \alpha) \right\|_F \left\| \frac{\partial \tilde{f}_m}{\partial w}(W, x_i; \alpha) - \frac{\partial \tilde{f}_m}{\partial w}(W(0), x_i; \alpha) \right\|_F \\
&\leq (\log m)^{2/\alpha} \left( \left\| \frac{\partial \tilde{f}_m}{\partial w}(W(0), x_i; \alpha) \right\|_F + \left\| \frac{\partial \tilde{f}_m}{\partial w}(W(0), x_i; \alpha) - \frac{\partial \tilde{f}_m}{\partial w}(W, x_i; \alpha) \right\|_F \right) \\
&\quad \times \left\| \frac{\partial \tilde{f}_m}{\partial w}(W, x_j; \alpha) - \frac{\partial \tilde{f}_m}{\partial w}(W(0), x_j; \alpha) \right\|_F \\
&\quad + (\log m)^{2/\alpha} \left\| \frac{\partial \tilde{f}_m}{\partial w}(W(0), x_j; \alpha) \right\|_F \left\| \frac{\partial \tilde{f}_m}{\partial w}(W, x_i; \alpha) - \frac{\partial \tilde{f}_m}{\partial w}(W(0), x_i; \alpha) \right\|_F.
\end{aligned}
$$

For every $i = 1, \ldots, k$,

$$
\begin{aligned}
(\log m)^{2/\alpha} \left\| \frac{\partial \tilde{f}_m}{\partial w}(W(0), x_i; \alpha) \right\|_F^2 &= \frac{1}{m^{2/\alpha}} \sum_{i=1}^m \langle w_i^{(0)}(0), x_i \rangle^2 I(|\langle w_i^{(0)}(0), x_i \rangle| > 0) \\
&\leq \frac{1}{m^{2/\alpha}} \sum_{i=1}^m \langle w_i^{(0)}(0), x_i \rangle^2,
\end{aligned}
$$

which converges in distribution, as $m \to \infty$. Thus there exist $M > 0$ and $m_0$ such that for every $m \geq m_0$ and every $i = 1, \ldots, k$,

$$
\mathbb{P}\left( (\log m)^{1/\alpha} \left\| \frac{\partial \tilde{f}_m}{\partial w}(W(0), x_i; \alpha) \right\|_F > M \right) < \frac{\delta}{8k^2}.
$$

By Lemma A.8, for $m$ sufficiently large, with probability at least $1 - \delta/(4k^2)$

$$
(\log m)^{1/\alpha} \left( \left\| \frac{\partial \tilde{f}_m}{\partial w}(W(0), x_i; \alpha) \right\|_F + \left\| \frac{\partial \tilde{f}_m}{\partial w}(W(0), x_i; \alpha) - \frac{\partial \tilde{f}_m}{\partial w}(W, x_i; \alpha) \right\|_F \right) < 2M
$$

whenever $\|W - W(0)\|_F < (\log m)^{2/\alpha}$. Lemma A.8 also implies that, for every $\gamma \in (0,1)$, and $i = 1, \ldots, k$, with probability at least $1 - \delta/(8k^2)$

$$
(\log m)^{1/\alpha} \left\| \frac{\partial \tilde{f}_m}{\partial w}(W, x_i; \alpha) - \frac{\partial \tilde{f}_m}{\partial w}(W(0), x_i; \alpha) \right\|_F < \frac{\lambda}{4Mk^2} m^{-\gamma/\alpha}
$$

whenever $\|W - W(0)\|_F^2 < (\log m)^{4/\alpha}$, provided $m$ is sufficiently large,. Thus, with probability at least $1 - \delta$, if $m$ is sufficiently large

$$
\max_{i,j} |\tilde{H}_m^{(2)}(W, X)[i,j] - \tilde{H}_m^{(2)}(W(0), X)[i,j]| < \frac{\lambda}{k^2} m^{-\gamma/\alpha},
$$

whenever $\|W - W(0)\|_F < (\log m)^{2/\alpha}$. Thus

$$
\begin{aligned}
&\|\tilde{H}_m^{(2)}(W, X) - \tilde{H}_m^{(2)}(W(0), X)\|_2 \\
&\leq \|\tilde{H}_m^{(2)}(W, X) - \tilde{H}_m^{(2)}(W(0), X)\|_F < \lambda m^{-\gamma/\alpha} < \frac{\lambda}{2},
\end{aligned}
$$

whenever $\|W - W(0)\|_F < (\log m)^{2/\alpha}$, provided $m$ is sufficiently large. The last inequality and Lemma A.6 imply that, with probability at least $1 - \delta$, if $m$ is sufficiently large, then

$$
\|\tilde{H}_m^{(2)}(W, X)\|_2 > \lambda/2,
$$

for every $W$ such that $\|W - W(0)\|_F < (\log m)^{2/\alpha}$. Since $\tilde{H}_m(W, X)$ is the sum of two positive semi-definite matrices $\tilde{H}_m^{(1)}(W, X)$ and $\tilde{H}_m^{(2)}(W, X)$, then

$$\|\tilde{H}_m(W, X)\|_2 \geq \|\tilde{H}_m^{(2)}(W, X)\|_2 > \lambda/2,$$

for every $W$ such that $\|W - W(0)\|_F < (\log m)^{2/\alpha}$, if $m$ is sufficiently large. $\qquad \square$

**Lemma A.10.** *For every $\delta > 0$ the following property holds true, for $m$ sufficiently large, with probabillity at least $1 - \delta$: there exists $M > 0$ such that*

$$(\log m)^{1/\alpha} \left\| \frac{\partial \tilde{f}_m}{\partial w^{(0)}}(W, x_j; \alpha) - \frac{\partial \tilde{f}_m}{\partial w^{(0)}}(W(0), x_j; \alpha) \right\|_F < M,$$

*for every $j = 1, \ldots, k$, and for every $W$ such that $\|W - W(0)\|_F \leq (\log m)^{2/\alpha}$.*

*Proof.* Let us define $r_i = |\langle w_i^{(0)} - w_i^{(0)}(0), x_j \rangle|$ for $i = 1, \ldots, m$. Now, since $\|x_j\| = 1$ by assumption, for $j = 1, \ldots, k$, then we can write

$$\sum_i r_i^2 \leq \|x_j\|^2 \cdot \|w_i^{(0)} - w^{(0)}(0)\|_F^2 \leq \|W - W(0)\|_F^2 \leq (\log m)^{4/\alpha}.$$

It holds

$$(\log m)^{2/\alpha} \left\| \frac{\partial \tilde{f}_m}{\partial w^{(0)}}(W, x_j; \alpha) - \frac{\partial \tilde{f}_m}{\partial w^{(0)}}(W(0), x_j; \alpha) \right\|_F^2$$

$$\leq \frac{1}{m^{2/\alpha}} \sum_{i=1}^{m} \left( w_i I(\langle w_i^{(0)}, x_j \rangle > 0) - w_i(0) I(\langle w_i^{(0)}(0), x_j \rangle > 0) \right)^2$$

$$\leq \frac{2}{m^{2/\alpha}} \sum_{i=1}^{m} (w_i - w_i(0))^2 I(\langle w_i^{(0)}, x_j \rangle > 0)$$

$$+ \frac{2}{m^{2/\alpha}} \sum_{i=1}^{m} w_i(0)^2 |I(\langle w_i^{(0)}, x_j \rangle > 0) - I(\langle w_i^{(0)}(0), x_j \rangle > 0)|.$$

We will bound the two terms separately. First,

$$\frac{2}{m^{2/\alpha}} \sum_{i=1}^{m} (w_i - w_i(0))^2 I(\langle w_i^{(0)}, x_j \rangle > 0)$$

$$\leq \frac{1}{m^{2/\alpha}} \sum_{i=1}^{m} (w_i - w_i(0))^2$$

$$\leq \frac{2}{m^{2(1-\gamma)/\alpha}} \|w - w(0)\|_F^2$$

$$\leq \frac{2}{m^{2/\alpha}} (\log m)^{4/\alpha} < \frac{c}{4} m^{-2\gamma/\alpha},$$

if $m$ is sufficiently large. To bound the second term, we can write that

$$\frac{2}{m^{2/\alpha}} \sum_{i=1}^{m} w_i(0)^2 |I(\langle w_i^{(0)}, x_j \rangle > 0) - I(\langle w_i^{(0)}(0), x_j \rangle > 0)|$$

$$\leq \frac{2}{m^{2/\alpha}} \sum_{i=1}^{m} w_i(0)^2,$$

which converges in distribution to a stable random variable, as $m \to \infty$. Hence there exists $M_1$ such that, with probability at least $1 - \delta/4$,

$$\frac{2}{m^{2/\alpha}} \sum_{i=1}^{m} (w_i - w_i(0))^2 I(\langle w_i^{(0)}, x_j \rangle > 0) < \frac{M_1^2}{2k^2}$$

and

$$\frac{2}{m^{2/\alpha}} \sum_{i=1}^{m} w_i(0)^2 |I(\langle w_i^{(0)}, x_j\rangle > 0) - I(\langle w_i^{(0)}(0), x_j\rangle > 0)| < \frac{M_1^2}{2k^2},$$

for $m$ sufficiently large, which entail

$$(\log m)^{1/\alpha} \left\| \frac{\partial \tilde{f}_m}{\partial w^{(0)}}(W, x_j; \alpha) - \frac{\partial \tilde{f}_m}{\partial w^{(0)}}(W(0)(\omega), x_j; \alpha) \right\|_F < \frac{M_1}{k}.$$

On the other hand, there exist $N_3 \in \mathcal{F}$ and $M_2$ with $P(N_3) > 1 - \delta/4$ such that, for every $\omega \in N_3$ and for $m$ sufficiently large,

$$\|\tilde{f}_m(W(0)(\omega), X; \alpha) - Y\|_F < M_2,$$

and

$$\max_{1 \leq i \leq k} \left\| \frac{\partial}{\partial W} \tilde{f}_m(W(0)(\omega), x_i; \alpha) \right\|_F < M_2 (\log m)^{-1/\alpha}.$$

The above inequalities follow from the convergence in distribution of $\tilde{f}_m(W(0), x_i; \alpha)$ and of

$$(\log m)^{2/\alpha} \left\| \frac{\partial}{\partial W} \tilde{f}_m(W(0), x_i; \alpha) \right\|_F^2 = \tilde{H}(W(0), X; \alpha)[i, i] \quad (i = 1, \dots, k),$$

as $m \to \infty$. $\qquad\qquad\qquad\qquad\qquad\qquad\qquad\qquad\qquad\qquad\qquad\qquad\qquad\qquad\qquad\qquad\qquad\qquad\quad$ $\square$

**Lemma A.11.** *Let $\gamma \in (0, 1)$ and $c > 0$ be fixed numbers. For every $\delta > 0$ the following property holds true, for $m$ sufficiently large, with probability at least $1 - \delta$:*

$$\|W(t) - W(0)\|_F < (\log m)^{2/\alpha}.$$

*if*

$$(\log m)^{2/\alpha} \left\| \frac{\partial \tilde{f}_m}{\partial w}(W(s), x_j; \alpha) - \frac{\partial \tilde{f}_m}{\partial w}(W(0), x_j; \alpha) \right\|_F^2 \leq c m^{-2\gamma/\alpha}$$

*for every NN's input $x_j$, with $j = 1, \dots, k$, and for every $s \leq t$.*

*Proof.* By Lemmas A.8 and A.9, there exists $N_1 \in \mathcal{F}$ with probability at least $1 - \delta/2$ such that, for every $\omega \in N_1$,

$$(\log m)^{2/\alpha} \left\| \frac{\partial \tilde{f}_m}{\partial w}(W, x_j; \alpha) - \frac{\partial \tilde{f}_m}{\partial w}(W(0)(\omega), x_j; \alpha) \right\|_F^2 < c m^{-2\gamma/\alpha},$$

for arbitrarily fixed $c >$ and $\gamma \in (0, 1/2)$, and

$$\lambda_{\min}(\tilde{H}_m(W, X)) > \frac{\lambda}{2},$$

for some $\lambda > 0$, for every $W$ such that $\|W - W(0)(\omega)\|_F \leq (\log m)^{2/\alpha}$ and every $j = 1, \dots, k$, provided $m$ is sufficiently large. Moreover, by Lemma A.10, there exist, for $m$ sufficiently large, $M_1 > 0$ and $N_2$ with $\mathbb{P}(N_2) > 1 - \delta$, such that

$$(\log m)^{1/\alpha} \left\| \frac{\partial \tilde{f}_m}{\partial w^{(0)}}(W, x_j; \alpha) - \frac{\partial \tilde{f}_m}{\partial w^{(0)}}(W(0)(\omega), x_j; \alpha) \right\|_F < \frac{M_1}{k},$$

for every $j = 1, \dots, k$, and for every $W$ such that $\|W - W(0)(\omega)\|_F \leq (\log m)^{2/\alpha}$. We will prove, by contradiction, that for every $\omega \in N_1 \cap N_2 \cap N_3$, $\|W(t) - W(0)\|_F < (\log m)^{2/\alpha}$ for every $t > 0$. In the following we will write $W(s)$ in the place of $W(s)(\omega)$ and always assume that $\omega$ belongs to $N_1 \cap N_2 \cap N_3$. Suppose that there exists $t$ such that $\|W(t) - W(0)\|_F \geq (\log m)^{2/\alpha}$, and let

$$t_0 = \operatorname{argmin}_{t \geq 0}\{t : \|W(t) - W(0)\|_F \geq (\log m)^{2/\alpha}\}.$$

Since $\|W(s) - W(0)\|_F \leq (\log m)^{2/\alpha}$ for every $s \leq t_0$, then, for every $s \leq t_0$,

$$\lambda_{\min}(\tilde{H}_m(W(s), X)) > \frac{\lambda}{2},$$

$$\left\|\frac{\partial \tilde{f}_m}{\partial w}(W(s), x_j; \alpha) - \frac{\partial \tilde{f}_m}{\partial w}(W(0), x_j; \alpha)\right\|_F < cm^{-\gamma/\alpha}(\log m)^{-1/\alpha},$$

$$\left\|\frac{\partial \tilde{f}_m}{\partial w^{(0)}}(W(s), x_j; \alpha) - \frac{\partial \tilde{f}_m}{\partial w^{(0)}}(W(0)(\omega), x_j; \alpha)\right\|_F < \frac{M_1}{k}(\log m)^{-1/\alpha} \quad (j = 1, \ldots, k),$$

$$\|\tilde{f}_m(W(0)(\omega), X; \alpha) - Y\|_F < M_2,$$

$$\max_{1 \leq i \leq k}\left\|\frac{\partial}{\partial W}\tilde{f}_m(W(0)(\omega), x_i; \alpha)\right\|_F < M_2(\log m)^{-1/\alpha}.$$

Let us now consider the gradient descent dynamic, with continuous learning rate $\eta = (\log m)^{2/\alpha}$:

$$\frac{\mathrm{d}W(s)}{\mathrm{d}s} = -(\log m)^{2/\alpha}\nabla_W \frac{1}{2}\sum_{i=1}^{k}\left(\tilde{f}_m(W(s), x_i; \alpha) - y_i\right)^2$$

$$= -(\log m)^{2/\alpha}\sum_{i=1}^{k}\left(\tilde{f}_m(W(s), x_i) - y_i\right)\frac{\partial \tilde{f}_m}{\partial W}(W(s), x_i; \alpha).$$

This expression allows to write

$$\|W(t_0) - W(0)\|_F$$

$$\leq \left\|\int_0^{t_0}\frac{\mathrm{d}}{\mathrm{d}s}W(s)\mathrm{d}s\right\|_F$$

$$\leq (\log m)^{2/\alpha}\left\|\int_0^{t_0}\sum_{i=1}^{k}(\tilde{f}_m(W(s), x_i; \alpha) - y_i)\frac{\partial \tilde{f}_m}{\partial W}(W(s), x_i; \alpha)\mathrm{d}s\right\|_F$$

$$\leq (\log m)^{2/\alpha}\max_{0 \leq s \leq t_0}\sum_{i=1}^{k}\left\|\frac{\partial \tilde{f}_m}{\partial W}(W(s), x_i; \alpha)\right\|_F \int_0^{t_0}\|\tilde{f}_m(W(s), X; \alpha) - Y\|\mathrm{d}s.$$

To bound the term $\|\tilde{f}_m(W(s), X; \alpha) - Y\|$ we will exploit the dynamics of the NN output

$$\frac{\mathrm{d}\tilde{f}_m(W(s), X; \alpha)}{\mathrm{d}s} = \frac{\partial \tilde{f}_m}{\partial W}(W(s), X; \alpha)\frac{\mathrm{d}W^T(s)}{\mathrm{d}s}$$

$$= -(\log m)^{2/\alpha}(\tilde{f}_m(W(s), X; \alpha) - Y)H_m(W(s), X)$$

$$= -(\tilde{f}_m(W(s), X; \alpha) - Y)\tilde{H}_m(W(s), X),$$

that gives

$$\frac{\mathrm{d}}{\mathrm{d}s}\|\tilde{f}_m(W(s), X; \alpha) - Y\|_2^2 = -2\left(\tilde{f}_m(W(s), X; \alpha) - Y\right)\tilde{H}_m(W(s), X)\left(\tilde{f}_m(W(s), X; \alpha) - Y\right)^T.$$

Since $\lambda_{\min}(\tilde{H}_m(W(s), X)) > \lambda/2$ for every $s \leq t_0$, then

$$\frac{\mathrm{d}}{\mathrm{d}s}\|\tilde{f}_m(W(s), X; \alpha) - Y\|_2^2 \leq -\lambda\|\tilde{f}_m(W(s), X; \alpha) - Y\|_2^2,$$

which implies that

$$\frac{\mathrm{d}}{\mathrm{d}s}\left(\exp(\lambda s)\|\tilde{f}_m(W(s), X; \alpha) - Y\|_2^2\right) \leq 0.$$

It follows that $\exp(\lambda s)\|\tilde{f}_m(W(s), X; \alpha) - Y\|_2^2$ is a decreasing function of $s$, and therefore

$$\|\tilde{f}_m(W(s), X; \alpha) - Y\|_2 \leq \exp(-\lambda/2)\|\tilde{f}_m(W(0), X; \alpha) - Y\|_2,$$

for every $s \leq t_0$. Substituting in the integral, we can write that

$$
\begin{aligned}
&\|W(t_0) - W(0)\|_F \\
&\leq (\log m)^{2/\alpha} \max_{0 \leq s \leq t_0} \sum_{i=1}^{k} \left\| \frac{\partial \tilde{f}_m}{\partial W}(W(s), x_i; \alpha) \right\|_F \int_0^{t_0} \exp(-\lambda s/2) \mathrm{d}s \cdot \|\tilde{f}_m(W(0), X; \alpha) - Y\| \\
&\leq \frac{2(\log m)^{2/\alpha}}{\lambda} \max_{0 \leq s \leq t_0} \sum_{i=1}^{k} \left( \left\| \frac{\partial \tilde{f}_m}{\partial W}(W(0), x_i; \alpha) \right\|_F + \left\| \frac{\partial \tilde{f}_m}{\partial W}(W(s), x_i; \alpha) - \frac{\partial \tilde{f}_m}{\partial W}(W(0), x_i; \alpha) \right\|_F \right) \\
&\quad \times \|\tilde{f}_m(W(0), X; \alpha) - Y\| \\
&\leq \frac{2(\log m)^{2/\alpha}}{\lambda} \max_{0 \leq s \leq t_0} \sum_{i=1}^{k} \left( \left\| \frac{\partial \tilde{f}_m}{\partial W}(W(0), x_i; \alpha) \right\|_F + \left\| \frac{\partial \tilde{f}_m}{\partial w^{(0)}}(W(s), x_i; \alpha) - \frac{\partial \tilde{f}_m}{\partial w^{(0)}}(W(0), x_i; \alpha) \right\|_F \right. \\
&\quad \left. + \left\| \frac{\partial \tilde{f}_m}{\partial w}(W(s), x_i; \alpha) - \frac{\partial \tilde{f}_m}{\partial w}(W(0), x_i; \alpha) \right\|_F \right) \|\tilde{f}_m(W(0), X; \alpha) - Y\| \\
&\leq \frac{2(\log m)^{1/\alpha}}{\lambda} \left( M_2 + M_1 + kcm^{-\gamma/\alpha} \right) M_2,
\end{aligned}
$$

which, for $m$ large, contradicts $\|W(t_0) - W(0)\|_F \geq (\log m)^{2/\alpha}$. $\qquad\square$

Now, we are in the position of proving Proposition 3.2. Let $m \in \mathbb{N}$ and $N \in \mathcal{F}$ be such that $\mathbb{P}(N) > 1 - \delta$ and the properties mentioned in Lemma A.8, Lemma A.9, Lemma A.10 and Lemma A.11 hold true for every $\omega \in N$. Therefore, by means of Lemma A.8 and of Lemma A.9, it is sufficient to show that

$$
\|W(t) - W(0)\|_F^2(\omega) < (\log m)^{2/\alpha}
$$

for every $t > 0$ and $\omega \in N$. By contradiction, suppose that there exists, for some $\omega \in N$, $t_0(\omega)$ finite with

$$
t_0(\omega) := \inf_{t \geq 0}\{t : \|W(t) - W(0)\|_F(\omega) \geq (\log m)^{2/\alpha}\}.
$$

Since $W(t)(\omega)$ is a continuous function of $t$, then $\|W(t_0(\omega)) - W(0)\|_F^2(\omega) = (\log m)^{2/\alpha}$. Then, by Lemma A.8,

$$
(\log m)^{2/\alpha} \left\| \frac{\partial \tilde{f}_m}{\partial W}(W(s), x_j; \alpha) - \frac{\partial \tilde{f}_m}{\partial W}(W(0), x_j; \alpha) \right\|_F^2 (\omega) < cm^{-2\gamma/\alpha},
$$

for every $s \leq t_0$ and every $j$. Therefore, by Lemma A.11 it holds true that $\|W(t_0(\omega)) - W(0)\|_F(\omega) < (\log m)^{2/\alpha}$, which contradicts the definition of $t_0$. This completes the proof of Proposition 3.2.

## B

The distribution of a random vector $\xi$ is said to be infinitely divisible if, for every $n$, there exist some i.i.d. random vectors $\xi_{n1}, \ldots, \xi_{nn}$ such that $\sum_k \xi_{nk} \overset{d}{=} \xi$. A $k$-dimensional random vector $\xi$ is infinitely divisible if and only if its characteristic function admits the representation $e^{\psi(u)}$, where

$$
\psi(u) = iu^T b - \frac{1}{2}u^T a u + \int \left( e^{iu^T x} - 1 - iu^T x I(\|x\| \leq 1) \right) \nu(dx) \tag{16}
$$

where $\nu$ is a measure on $\mathbb{R}^k \setminus \{0\}$ satisfying $\int (\|x\|^2 \wedge 1)\nu(dx) < \infty$, $a$ is a $k \times k$ positive semi-definite, symmetric matrix and $b$ is a vector. The measure $\nu$ is called the Lévy measure of $\xi$ and $(a, b, \nu)$ are called the characteristics of the infinitely divisible distribution. We will write $\xi \sim i.d.(a, b, \nu)$. Other kinds of truncation can be used for the term $iu^T x$. This affects only the vector of centering constants $b$. An i.i.d. array of random vectors is a collection of random vectors $\{\xi_{nj}, j \leq m_n, n \geq 1\}$ such that, for every $n$,

$\xi_{n1}, \ldots, \xi_{nm_n}$ are i.i.d. The class of infinitely divisible distributions coincides with the class of limits of sums of i.i.d. arrays (Kallenberg, 2002, Theorem 13.12).

To state a general criterion of convergence, we first introduce some notations. Let $\xi \sim i.d.(a, b, \nu)$. Define, for each $h > 0$,

$$a^{(h)} = a + \int_{||x|| < h} xx^T \nu(dx),$$

$$b^{(h)} = b - \int_{h < ||x|| \leq 1} x\nu(dx),$$

where $\int_{h < ||x|| \leq 1} = -\int_{1 < ||x|| \leq h}$ if $h > 1$. Denote by $\xrightarrow{v}$ vague convergence, that is convergence of measures with respect to the topology induced by bounded, measurable functions with compact support. Moreover, let $\overline{\mathbb{R}^k}$ be the one-point compactification of $\mathbb{R}^k$. The following criterion for convergence holds (Kallenberg, 2002, Corollary 13.16).

**Theorem B.1.** *Consider in $\mathbb{R}^k$ an i.i.d. array $(\xi_{nj})_{j=1,\ldots,m_n, n \geq 1}$ and let $\xi$ be i.d.$(a, b, \nu)$. Let $h > 0$ be such that $\nu(||x|| = h) = 0$. Then $\sum_j \xi_{nj} \xrightarrow{d} \xi$ if and only if the following conditions hold:*

*(i)* $m_n \mathbb{P}(\xi_{n1} \in \cdot) \xrightarrow{v} \nu(\cdot)$ *on* $\overline{\mathbb{R}^k} \setminus \{0\}$

*(ii)* $m_n \mathbb{E}(\xi_{n1} \xi_{n1}^T I(||\xi_{n1}|| < h)) \to a^{(h)}$

*(iii)* $m_n \mathbb{E}(\xi_{n1} I(||\xi_{n1}|| < h)) \to b^{(h)}$

Inside the class of infinitely divisible distribution, we can distinguish the subclass of stable distributions. A $k$-dimensional random vector $\xi$ has stable distribution if, for every independent random vectors $\xi_1$ and $\xi_2$ with $\xi_1 \stackrel{d}{=} \xi_2 \stackrel{d}{=} \xi$ and every $a, b \in \mathbb{R}$, there exists $c \in \mathbb{R}$ and $d \in \mathbb{R}^k$ such that $a\xi_1 + b\xi_2 \stackrel{d}{=} c\xi + d$. This is equivalent to the condition: for every $n \geq 1$,

$$\xi_1 + \cdots + \xi_n \stackrel{d}{=} n^{1/\alpha}\xi + d_n \tag{17}$$

where $\alpha \in (0, 2]$, $\xi_1, \ldots, \xi_n$ are i.i.d. copies of $\xi$ and $d_n$ is a vector. The random vector $\xi$ is said to be strictly stable if (17) holds with $d_n = 0$. A stable vector $\xi$ is strictly stable if and only if all its components are strictly stable. The coefficient $\alpha$ is called the index of stability of $\xi$ and the law of $\xi$ is called $\alpha$-stable. A stable vector $\xi$ is symmetric stable if $\mathbb{P}(\xi \in A) = \mathbb{P}(-\xi \in A)$ for every Borel set $A$. A symmetric stable vector is strictly stable. The class of stable distributions coincides with the class of limit laws of sequences $((\sum_{k=1}^n X_k - b_n)/a_n)$, where $(X_n)$ are i.i.d. random variables.

A stable distribution is infinitely divisible. Thus its characteristic function admits the Lévy representation (16). If $\alpha = 2$, then the Lévy measure is the null measure and, therefore, the stable distribution coincides with the multivariate normal distribution with covariance matrix $a$ and mean vector $b$. If $\alpha < 2$, then $a = 0$ (the zero matrix) and the $\alpha$-stability implies that there exists a measure $\sigma$ on the unit sphere $\mathbb{S}^{k-1}$ such that $\nu(dx) = r^{-(\alpha+1)} dr\sigma(ds)$, where $r = ||x||$ and $s = x/||x||$. Substituting in (16), we obtain

$$\psi(u) = iu^T b + \int_S \int_0^\infty \left( e^{iru^T s} - 1 - iru^T s I(r \leq 1) \right) \frac{1}{r^{1+\alpha}} dr\sigma(ds)$$

For $\alpha < 1$, the centering $iru^T s I(r \leq 1)$ is not needed, since the function (of $r$) is integrable, and we can write

$$\psi(u) = iu^T b' + \int_S \int_0^\infty \left( e^{iru^T s} - 1 \right) \frac{1}{r^{1+\alpha}} dr\sigma(ds),$$

for some vector $b'$. After evaluating the inner integrals as in Feller (1968, Example XVII.3), we obtain

$$\psi(u) = iu^T b' - \int_S |u^T s|^\alpha \Gamma(1 - \alpha) \left( \cos(\pi\alpha/2) - i \, \mathrm{sign}(u^T s) \sin(\pi\alpha/2) \right) \sigma(ds)$$

$$= iu^T b' - \int_S |u^T s|^\alpha \left(1 - i\,\text{sign}(u^T s)\tan(\pi\alpha/2)\right)\Gamma(1-\alpha)\cos(\pi\alpha/2)\sigma(ds).$$

For $\alpha > 1$, using the centering $iru^T s$, we can write

$$\psi(u) = iu^T b'' + \int_S \int_0^\infty \left(e^{iru^T s} - 1 - iru^T s\right)\frac{1}{r^{1+\alpha}}dr\sigma(ds),$$

for some $b''$. After evaluating the inner integrals as in Feller (1968, Example XVII.3), we obtain

$$\psi(u) = iu^T b'' + \int_S |u^T s|^\alpha \frac{\Gamma(2-\alpha)}{\alpha-1}\left(\cos(\pi\alpha/2) - i\,\text{sign}(u^T s)\sin(\pi\alpha/2)\right)\sigma(ds)$$

$$= iu^T b'' - \int_S |u^T s^\alpha \left(1 - i\,\text{sign}(u^T s)\tan(\pi\alpha/2)\right)\frac{\Gamma(2-\alpha)}{1-\alpha}\cos(\pi\alpha/2)\sigma(ds).$$

Since, for $\alpha < 1$, $\Gamma(2-\alpha) = (1-\alpha)\Gamma(1-\alpha)$, we can encompass the above results in one equation, and write, for $\alpha \neq 1$,

$$\psi(u) = iu^T b''' - \int_S |u^T s|^\alpha \left(1 - i\,\text{sign}(u^T s)\tan(\pi\alpha/2)\right)\frac{\Gamma(2-\alpha)}{1-\alpha}\cos(\pi\alpha/2)\sigma(ds),$$

for some $b'''$. Finally, for $\alpha = 1$, using the centering $ir\sin ru^T s$, we can write

$$\psi(u) = iu^T b'''' + \int_S \int_0^\infty \left(e^{iru^T s} - 1 - ir\sin ru^T s\right)\frac{1}{r^2}dr\sigma(ds),$$

for some $b''''$. Evaluating the inner integral as in Feller (1968, Example XVII.3), we obtain

$$\psi(u) = iu^T b'''' - \int_S |u^T s|\left(\frac{\pi}{2} + i\text{sign}(u^T s)\log|u^T s|\right)\sigma(ds)$$

$$= iu^T b'''' - \int_S |u^T s|\left(1 + i\frac{2}{\pi}\text{sign}(u^T s)\log|u^T s|\right)\frac{\pi}{2}\sigma(ds).$$

Considering the spectral representation $e^{\psi(u)}$ of the multivariate stable characteristic function

$$\psi(u) = \begin{cases} -\int_S |u^T s|^\alpha \left(1 - i\,\text{sign}(u^T s)\tan(\pi\alpha/2)\right)\Gamma(ds) + iu^T \mu^{(0)} & \alpha \neq 1 \\ -\int_S |u^T s|\left(1 + i\frac{2}{\pi}\,\text{sign}(u^T s)\log|u^T s|\right)\Gamma(ds) + iu^T \mu^{(0)} & \alpha = 1, \end{cases}$$

we can establish the following relationship between the Lévy measure $\nu$ and the spectral measure $\Gamma$:

$$\nu(dx) = C_\alpha \frac{1}{r^{\alpha+1}}\Gamma(ds),$$

where $r = ||x||$, $s = x/||x||$ and

$$C_\alpha = \begin{cases} \dfrac{1-\alpha}{\Gamma(2-\alpha)\cos(\pi\alpha/2)} & \alpha \neq 1 \\ \\ 2/\pi & \alpha = 1 \end{cases}$$

A Stable random vector $\xi$ is strictly stable if and only if

$$\begin{cases} \mu^{(0)} = 0 & \alpha \neq 1 \\ \int_S s_j\Gamma(ds) = 0 \text{ for every j} & \alpha = 1. \end{cases}$$

(see e.g. Samoradnitsky and Taqqu (1994, Theorem 2.4.1)). By Theorem B.1, the spectral measure $\Gamma$ of a symmetric stable random vector $\xi$ satisfies

$$\lim_{n\to\infty} n\mathbb{P}\left(||\xi|| > n^{1/\alpha}x, \frac{\xi}{||\xi||} \in A\right) = C_\alpha x^{-\alpha}\Gamma(A) \tag{18}$$

for every Borel set $A$ of $S$ such that $\Gamma(\partial A) = 0$. Moreover, the distribution of a random vector $\xi$ belongs to the domain of attraction of the $\text{St}_k(\alpha, \Gamma)$ distribution, with $\alpha \in (0,2)$ and $\Gamma$ simmetric finite measure on $\mathbb{S}^{k-1}$, if and only if (18) holds (see e.g. Davydov et al. (2008, Theorem 4.3)).

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
