# OpenReview forum: "Large-width asymptotics and training dynamics of $\alpha$-Stable ReLU neural networks"
_TMLR — Accepted by TMLR_

### Review · Reviewer_KFEa · 2024-06-12

**Summary Of Contributions:**

This paper considers the properties of large width neural networks using ReLU activation functions and initial weight distribution sampled from an $\alpha$-stable distribution. The paper shows that the (re-scaled) weights kernel defined by the initial distribution (synonymous with Du et al's neural tagent kernel) converges to another $\alpha$-stable distribution. From this result, the paper then is able to show that such a network almost-surely converges to zero training error when optimizing a squared loss using gradient descent.

**Audience:**

Yes

**Broader Impact Concerns:**

I have no concerns about this paper's ethical implications.

**Claims And Evidence:**

Yes

**Requested Changes:**

I highly suggest re-writing this paper for clarity, and thinking about how to address the examples provided above.

One critical aspect which needs fixing for this paper to be publishable is the references section. It is inconsistent in what names it uses for authors, and whether the first or last name appears first in the entry making searching through it difficult. Also some of the papers listed as arxiv publications have full publications one example is Lee et al 2022 (which has now been published at JMLR). I did not check all entries.

While I did spot check some of the proofs, I do not have the expertise to validate their accuracy. The results seem reasonable, but I will hold judgment until I see other reviews.

**Strengths And Weaknesses:**

## Strengths

- The main result is a clear hole in the literature, and is a "straightforward" extension of several important results to this new setting. (I use straightforward not in the sense that the proofs are straightforward, but in the sense it is clear how these results could be extending previous results).

- The contributions are clearly laid out early, and connections to the literature are clearly made throughout the paper.

## Weaknesses

The primary weakness for this paper is in the presentation. I often found it difficult to piece together exactly how the different theorems were fitting together to provide evidence for the final claim (i.e. theorem 4.4). This is not to say all the ideas were not presented, but instead the ideas were not presented clearly enough without considerable work from the reader to parse the different sections.

I will go through some examples on how this could be improved.

- E1: The introduction dives right into the mathematical background and setting of prior works. While I appreciate the background, you delay the motivation on why this work is important until later.

- E2: The introduction has several in-line lists with the assumptions/constraints of prior work. These lists are often hard to parse and relate to the result you are considering. It might be worth splitting these lists out and using more space to go through these ideas (you have plenty of room still).

- E3: The motivation behind using $\alpha$-stable distributions is buried behind awkward dense language. It isn't until part way through the first paragraph of section 1.1 did I begin to understand why we want to use these distributions (i.e. the long-tail effects allow for potentially better representational qualities).

- E4: You use the ability represent "hidden features" as main motivation for the $\alpha$-stable distributions, but what this means is unclear without going through the prior literature. You should contextualize this in more straight forward language. Lee et al 2019 do a better job by providing the full quote from Neal 1995. This likely comes from me not understanding what is meant by $\alpha$-stable NNs being "more flexible" than Gaussian NNs.

- E5: Your contributions section is too dense and does not just highlight your contributions. Here you should precisely enumerate your contributions and link to the appropriate sections/theorems. While they don't enumerate, Arora et al, 2019 does an excellent job at clearly stating their contributions and the impact of these contributions in plain language.

While I provide only 5 examples, the style throughout is overly dense in my opinion. The technical content is unavoidable (and shouldn't be removed), but taking inspiration from [Lee et al, 2019], [Arora et al, 2019], and [Du et al, 2019] would improve the quality of the paper and improve its readability. Some notable properties of their style:
- Taking room for lists (not in-lining lists multiple times)
- Giving more plain language descriptions throughout, and saving the precise mathematics for where they are needed.
- Paragraphs with clear purpose. This paper has many large paragraphs that often meandered through several topics.


Questions:

- Q1: What are the consequences of fluctuations of H_m^(1) during training?
- Q2: What interesting phenomena do you mean in the last sentence of the first paragraph of your discussion? Are these phenomena your results or do you mean more generally? Some insights here would be great, because I appreciate the theoretical consequences, I'm struggling to understand how the use of $\alpha$-stable distributions relate back to practice.
- Q3: Can we compare the convergence rates of $\alpha$-stable networks to their gaussian counterparts? I think both are linear, but maybe there isn't much we can say about the absolute values given how this was proved.  Maybe this is in the paper, but I missed it.
- Q4: Why is $\alpha=2$ excluded from your analysis? How difficult would it be to include? What results are valid for $\alpha=2$?

This paper seems relevant and is not cited:
- "alpha-Stable convergence of heavy-tailed infinitely-wide neural networks", Paul Jung et al 2021

Some minor edits I saw:
- Sentence 1 of the introduction "namely NNs with weights *that* are Gaussian distributed".
- In the first equation of the introduction should it be $w_{ij} instead of just $w_i$?

---

### Review · Reviewer_yfNq · 2024-07-19

**Summary Of Contributions:**

The paper analyzes large-width asymptotics and training dynamics of $\alpha$-stable ReLU neural networks (NNs). The main contributions are as follows,

i)  The prior results showed Gaussian NNs, and their generalization $\alpha$-stable NNs converge weakly to the Gaussian and $\alpha$-stable process respectively. On this front, the contribution of the paper is an extension of the results on $\alpha$-stable NNs to $\alpha$-stable ReLU NNs. The results suggest the choice of activation function affects the scaling of the NN to achieve the infinitely wide $\alpha$-stable process. Authors show that ReLU activation scaling by $(m \log m)^{-1/\alpha}$ is required,  in contrast to prior works on other activation functions where $m^{-1/\alpha}$ scaling was required.

ii)  Authors characterize the training dynamics of $\alpha$-stable ReLU NNs in terms of a random kernel they refer to as the $\alpha$-stable NTK and show when such a network with sufficiently large width is training using gradient descent then zero training error is achieved at a linear rate. These results generalize prior results on understanding the gradient descent’s training dynamics on Gaussian NNs (Jacot et al. 2018, Arora et al. 2019, Du et al. 2019, Lee et al. 2019).

**Audience:**

Yes

**Claims And Evidence:**

Yes

**Requested Changes:**

1. Please walk through the reader in sections 3 and 4. In 3, can you explain in words why the factor of  $(m \log m)^{-1/\alpha}$ is required? The previous results have shown the convergence of $\alpha$-stable NNs with other activations, what makes results with ReLU interesting and what is the main challenge that is introduced due to ReLU and how do the authors tackle it? Please try to answer these questions in section 3 without going into too much notation and algebra. I’d like to see similar changes in section 4. Please start the section with an overview of what you want to show and a rough overview of how you plan to prove those results.

2. It might be helpful to have informal theorem statements for the main results in the contributions section.

3. Please include simulations on the $\alpha-$Stable ReLU NNs with some choices of $alpha$ (distributions), i) to show the convergence to $\alpha-$Stable process with infinite width ii) to validate the claims made on the training dynamics of these NNs using gradient descent.

**Strengths And Weaknesses:**

### Strengths

1. The paper extends the characterization of the large-width distribution of $\alpha-$Stable $\phi-$NNs to the ReLU activation function and generalizes previous results to the $\alpha-$Stable setting.

2. The authors analyze the dynamics of training such networks using gradient descent. They first show in the infinite width setting the kernel converges to the so-called $\alpha-$Stable Neural Tangent Kernel (NTK) a positive definite random kernel.

3. The paper provides convergence guarantees for the gradient descent algorithm, showing that it achieves zero training loss at a linear rate, with high probability, even in the presence of randomness in the $\alpha-$Stable NTK.


### Weaknesses
1. It would make a stronger case with some simulations validating the theoretical results.

2. The paper focuses only on the math and the technical aspects of the proofs (some of which could be deferred to the appendix), without providing any walkthroughs, intuitions, and discussions, especially in sections 3 and 4.

---

### Review · Reviewer_Szs9 · 2024-07-31

**Summary Of Contributions:**

This paper studies the gradient dynamics of $\alpha$-stable ReLU neural networks when the neuron neural width reaches infinity.

**Audience:**

Yes

**Claims And Evidence:**

Yes

**Requested Changes:**

See weakness

**Strengths And Weaknesses:**

Strength:

1. This paper provides a solid theoretical framework.
2. A detailed description of related works, especially for NTK.

Weakness:

1. It is unclear to me what the new insights from this paper are compared with existing NTK frameworks.

2. The assumptions still follow those from NTK, which are neither practical nor novel.

---

### Decision · Action_Editor_cYYR · 2024-10-03

**Recommendation:** Accept with minor revision

**Comment:**

This paper received positive reviews however it did not receive strong support from the reviewers.

Reviewers have requested certain edits; the authors should carefully go over those and address them in their final version of the paper. Note that any questions asked by reviewers should be clarified in the main text as well.

**Audience:**

Yes, the paper focuses on the training dynamics of ReLU neural networks. There is audience for this kind of work among TMLR community.

**Claims And Evidence:**

Yes, the claims in this paper are supported by theoretical results and rigorous proofs.